# Efficient Probabilistic Logic Reasoning with Graph Neural Networks

**Yuyu Zhang**[1]**, Xinshi Chen**[1]**, Yuan Yang**[1]**, Arun Ramamurthy**[2]**,**
**Bo Li**[3]**, Yuan Qi**[4] **& Le Song**[1,4]
[1]Georgia Institute of Technology   [2]Siemens Corporate Technology
[3]University of Illinois at Urbana Champaign   [4]Ant Financial
`{yuyu,xinshi.chen,yyang754}@gatech.edu`
`arun.ramamurthy@siemens.com,lbo@illinois.edu`
`yuan.qi@antfin.com,lsong@cc.gatech.edu`

## Abstract

Markov Logic Networks (MLNs), which elegantly combine logic rules and probabilistic graphical models, can be used to address many knowledge graph problems. However, inference in MLN is computationally intensive, making the industrial-scale application of MLN very difficult. In recent years, graph neural networks (GNNs) have emerged as efficient and effective tools for large-scale graph problems. Nevertheless, GNNs do not explicitly incorporate prior logic rules into the models, and may require many labeled examples for a target task. In this paper, we explore the combination of MLNs and GNNs, and use graph neural networks for variational inference in MLN. We propose a GNN variant, named ExpressGNN, which strikes a nice balance between the representation power and the simplicity of the model. Our extensive experiments on several benchmark datasets demonstrate that ExpressGNN leads to effective and efficient probabilistic logic reasoning.

## 1 Introduction

Knowledge graphs collect and organize relations and attributes about entities, which are playing an increasingly important role in many applications, including question answering and information retrieval. Since knowledge graphs may contain incorrect, incomplete or duplicated records, additional processing such as link prediction, attribute classification, and record de-duplication is typically needed to improve the quality of knowledge graphs and derive new facts.

Markov Logic Networks (MLNs) were proposed to combine hard logic rules and probabilistic graphical models, which can be applied to various tasks on knowledge graphs (Richardson & Domingos, 2006). The logic rules incorporate prior knowledge and allow MLNs to generalize in tasks with small amount of labeled data, while the graphical model formalism provides a principled framework for dealing with uncertainty in data. However, inference in MLN is computationally intensive, typically exponential in the number of entities, limiting the real-world application of MLN. Also, logic rules can only cover a small part of the possible combinations of knowledge graph relations, hence limiting the application of models that are purely based on logic rules.

Graph neural networks (GNNs) have recently gained increasing popularity for addressing many graph related problems effectively (Dai et al., 2016; Li et al., 2016; Kipf & Welling, 2017; Schlichtkrull et al., 2018). GNN-based methods typically require sufficient labeled instances on specific end tasks to achieve good performance, however, knowledge graphs have the long-tail nature (Xiong et al., 2018), i.e., a large portion the relations in only are a few triples. Such data scarcity problem among long-tail relations poses tough challenge for purely data-driven methods.

In this paper, we explore the combination of the best of both worlds, aiming for a method which is data-driven yet can still exploit the prior knowledge encoded in logic rules. To this end, we design a simple variant of graph neural networks, named ExpressGNN, which can be efficiently trained in the variational EM framework for MLN. An overview of our method is illustrated in Fig. 1. ExpressGNN and the corresponding reasoning framework lead to the following desiderata:

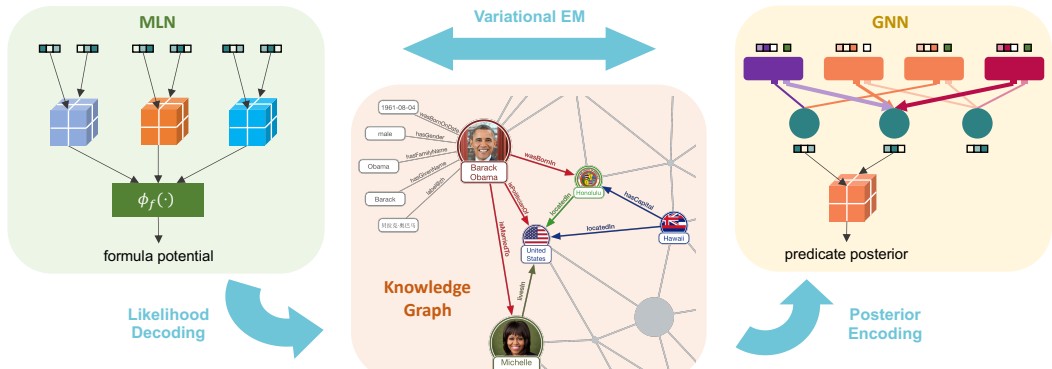

Figure 1: Overview of our method for combining MLN and GNN using the variational EM framework.

- *Efficient inference and learning:* ExpressGNN can be viewed as the inference network for MLN, which scales up MLN inference to much larger knowledge graph problems.

- *Combining logic rules and data supervision:* ExpressGNN can leverage the prior knowledge encoded in logic rules, as well as the supervision from graph structured data.

- *Compact and expressive model:* ExpressGNN may have small number of parameters, yet it is sufficient to represent mean-field distributions in MLN.

- *Capability of zero-shot learning:* ExpressGNN can deal with the zero-shot learning problem where the target predicate has few or zero labeled instances.

## 2 RELATED WORK

**Statistical relational learning.** There is an extensive literature relating the topic of logic reasoning. Here we only focus on the approaches that are most relevant to statistical relational learning on knowledge graphs. Logic rules can compactly encode the domain knowledge and complex dependencies. Thus, hard logic rules are widely used for reasoning in earlier attempts, such as expert systems (Ignizio, 1991) and inductive logic programming (Muggleton & De Raedt, 1994). However, hard logic is very brittle and has difficulty in coping with uncertainty in both the logic rules and the facts in knowledge graphs. Later studies have explored to introduce probabilistic graphical model in logic reasoning, seeking to combine the advantages of relational and probabilistic approaches. Representative works including Relational Markov Networks (RMNs; Taskar et al. (2007)) and Markov Logic Networks (MLNs; Richardson & Domingos (2006)) were proposed in this background.

**Markov Logic Networks.** MLNs have been widely studied due to the principled probabilistic model and effectiveness in a variety of reasoning tasks, including entity resolution (Singla & Domingos, 2006a), social networks (Zhang et al., 2014), information extraction (Poon & Domingos, 2007), etc. MLNs elegantly handle the noise in both logic rules and knowledge graphs. However, the inference and learning in MLNs is computationally expensive due to the exponential cost of constructing the ground Markov network and the NP-complete optimization problem. This hinders MLNs to be applied to industry-scale applications. Many works appear in the literature to improve the original MLNs in both accuracy (Singla & Domingos, 2005; Mihalkova & Mooney, 2007) and efficiency (Singla & Domingos, 2006b; 2008; Poon & Domingos, 2006; Khot et al., 2011; Bach et al., 2015). Nevertheless, to date, MLNs still struggle to handle large-scale knowledge bases in practice. Our framework ExpressGNN overcomes the scalability challenge of MLNs by efficient stochastic training algorithm and compact posterior parameterization with graph neural networks.

**Graph neural networks.** Graph neural networks (GNNs; Dai et al. (2016); Kipf & Welling (2017)) can learn effective representations of nodes by encoding local graph structures and node attributes. Due to the compactness of model and the capability of inductive learning, GNNs are widely used in modeling relational data (Schlichtkrull et al., 2018; Battaglia et al., 2018). Recently, Qu et al. (2019) proposed Graph Markov Neural Networks (GMNNs), which employs GNNs together with conditional random fields to learn object representations. These existing works are simply data-driven, and not able to leverage the domain knowledge or human prior encoded in logic rules. To the best

of our knowledge, ExpressGNN is the first work that connects GNNs with first-order logic rules to combine the advantages of both worlds.

**Knowledge graph embedding.** Another line of research for knowledge graph reasoning is in the family of knowledge graph embedding methods, such as TransE (Bordes et al., 2013), NTN (Socher et al., 2013), DistMult (Kadlec et al., 2017), ComplEx (Trouillon et al., 2016), and RotatE (Sun et al., 2019). These methods design various scoring functions to model relational patterns for knowledge graph reasoning, which are very effective in learning the transductive embeddings of both entities and relations. However, these methods are not able to leverage logic rules, which can be crucial in some relational learning tasks, and have no consistent probabilistic model. Compared to these methods, ExpressGNN has consistent probabilistic model built in the framework, and can incorporate knowledge from logic rules. A recent concurrent work Qu & Tang (2019) has proposed probabilistic Logic Neural Network (pLogicNet), which integrates knowledge graph embedding methods with MLNs with EM framework. Compared to pLogicNet which uses a flattened embedding table as the entity representation, our work explicitly captures the structure knowledge encoded in the knowledge graph with GNNs and supplement the knowledge from logic formulae for the prediction task.

# 3 PRELIMINARY

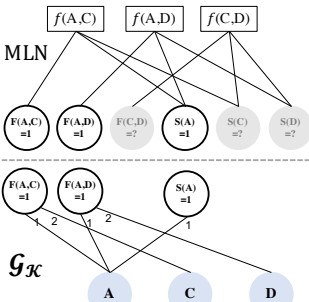

**Knowledge Graph.** A knowledge graph is a tuple $\mathcal{K} = (\mathcal{C}, \mathcal{R}, \mathcal{O})$ consisting of a set $\mathcal{C} = \{c_1, \ldots, c_M\}$ of $M$ entities, a set $\mathcal{R} = \{r_1, \ldots, r_N\}$ of $N$ relations, and a collection $\mathcal{O} = \{o_1, \ldots, o_L\}$ of $L$ observed facts. In the language of first-order logic, entities are also called constants. For instance, a constant can be a person or an object. Relations are also called **predicates**. Each predicate is a logic function defined over $\mathcal{C}$, i.e., $r(\cdot) : \mathcal{C} \times \ldots \times \mathcal{C} \mapsto \{0, 1\}$. In general, the arguments of predicates are asymmetric. For instance, for the predicate $r(c, c') := \text{L}(c, c')$ (L for Like) which checks whether $c$ likes $c'$, the arguments $c$ and $c'$ are not exchangeable.

Figure 2: *Bottom*: A knowledge base as a factor graph. $\{A, C, D\}$ are entities, and F (Friend) and S (Smoke) are predicates. *Top*: Markov Logic Network (MLN) with formula $f(c, c') := \neg \text{S}(c) \vee \neg \text{F}(c, c') \vee \text{S}(c')$. Shaded circles correspond to latent variables.

With a particular set of entities assigned to the arguments, the predicate is called a ground predicate, and **each ground predicate $\equiv$ a binary random variable**, which will be used to define MLN. For a $d$-ary predicate, there are $M^d$ ways to ground it. We denote an assignment as $a_r$. For instance, with $a_r = (c, c')$, we can simply write a ground predicate $r(c, c')$ as $r(a_r)$. Each observed fact in knowledge bases is a truth value $\{0, 1\}$ assigned to a ground predicate. For instance, a fact $o$ can be $[\text{L}(c, c') = 1]$. The number of observed facts is typically much smaller than that of unobserved facts. We adopt the open-world paradigm and treat these **unobserved facts $\equiv$ latent variables**.

As a clearer representation, we express a knowledge base $\mathcal{K}$ by a bipartite graph $\mathcal{G}_{\mathcal{K}} = (\mathcal{C}, \mathcal{O}, \mathcal{E})$, where nodes on one side of the graph correspond to constants $\mathcal{C}$ and nodes on the other side correspond to observed facts $\mathcal{O}$, which is called *factor* in this case. The set of $T$ edges, $\mathcal{E} = \{e_1, \ldots, e_T\}$, connect constants and the observed facts. More specifically, an edge $e = (c, o, i)$ between node $c$ and $o$ exists, if the ground predicate associated with $o$ uses $c$ as an argument in its $i$-th argument position (Fig. 2).

**Markov Logic Networks.** MLNs use logic formulae to define potential functions in undirected graphical models. A logic formula $f(\cdot) : \mathcal{C} \times \ldots \times \mathcal{C} \mapsto \{0, 1\}$ is a binary function defined via the composition of a few predicates. For instance, a logic formula $f(c, c')$ can be

$\text{Smoke}(c) \wedge \text{Friend}(c, c') \Rightarrow \text{Smoke}(c') \iff \neg \text{Smoke}(c) \vee \neg \text{Friend}(c, c') \vee \text{Smoke}(c'),$

where $\neg$ is negation and the equivalence is established by De Morgan's law. Similar to predicates, we denote an assignment of constants to the arguments of a formula $f$ as $a_f$, and the entire collection of consistent assignments of constants as $\mathcal{A}_f = \{a_f^1, a_f^2, \ldots\}$. A formula with constants assigned to all of its arguments is called a ground formula. Given these logic representations, MLN can be defined as a joint distribution over all observed facts $\mathcal{O}$ and unobserved facts $\mathcal{H}$ as

$$P_w(\mathcal{O}, \mathcal{H}) := \tfrac{1}{Z(w)} \exp \Big( \textstyle\sum_{f \in \mathcal{F}} w_f \sum_{a_f \in \mathcal{A}_f} \phi_f(a_f) \Big), \tag{1}$$

where $Z(w)$ is the partition function summing over all ground predicates and $\phi_f(\cdot)$ is the potential function defined by a formula $f$ as illustrated in Fig. 2. One form of $\phi_f(\cdot)$ can simply be the truth

value of the logic formula $f$. For instance, if the formula is $f(c, c') := \neg S(c) \vee \neg F(c, c') \vee S(c')$, then $\phi_f(c, c')$ can simply take value 1 when $f(c, c')$ is true and 0 otherwise. Other more sophisticated $\phi_f$ can also be designed, which have the potential to take into account complex entities, such as images or texts, but will not be the focus of this paper. The weight $w_f$ can be viewed as the confidence score of the formula $f$: the higher the weight, the more accurate the formula is.

**Difference between KG and MLN.** We note that the graph topology of knowledge graphs and MLN can are very different, although MLN is defined on top of knowledge graphs. Knowledge graphs are typically very sparse, where the number of edges (observed relations) is typically linear in the number of entities. However, the graphs associated with MLN are much denser, where the number of nodes can be quadratic or more in the number of entities, and the number of edges (dependency between variables) is also high-order polynomials in the number of entities.

# 4 VARIATIONAL EM FOR MARKOV LOGIC NETWORKS

In this section, we introduce the variational EM framework for MLN inference and learning, where we will use ExpressGNN as a key component (detailed in Sec. 5). Markov Logic Networks model the joint probabilistic distribution of all observed and latent variables, as defined in Eq. 1. This model can be trained by maximizing the log-likelihood of all the observed facts $\log P_w(\mathcal{O})$. However, it is intractable to directly maximize the objective, since it requires to compute the partition function $Z(w)$ and integrate over all variables $\mathcal{O}$ and $\mathcal{H}$. We instead optimize the variational evidence lower bound (ELBO) of the data log-likelihood, as follows

$$\log P_w(\mathcal{O}) \geqslant \mathcal{L}_{\text{ELBO}}(Q_\theta, P_w) := \mathbb{E}_{Q_\theta(\mathcal{H}|\mathcal{O})}\Big[\log P_w(\mathcal{O}, \mathcal{H})\Big] - \mathbb{E}_{Q_\theta(\mathcal{H}|\mathcal{O})}\Big[\log Q_\theta(\mathcal{H}|\mathcal{O})\Big], \quad (2)$$

where $Q_\theta(\mathcal{H} \mid \mathcal{O})$ is a variational posterior distribution of the latent variables given the observed ones. The equality in Eq. 2 holds if the variational posterior $Q_\theta(\mathcal{H}|\mathcal{O})$ equals to the true posterior $P_w(\mathcal{H}|\mathcal{O})$. We then use the variational EM algorithm (Ghahramani et al., 2000) to effectively optimize the ELBO. The variational EM algorithm consists of an expectation step (E-step) and a maximization step (M-step), which will be called in an alternating fashion to train the model: 1) In the E-step (Sec. 4.1), we infer the posterior distribution of the latent variables, where $P_w$ is fixed and $Q_\theta$ is optimized to minimize the KL divergence between $Q_\theta(\mathcal{H}|\mathcal{O})$ and $P_w(\mathcal{H}|\mathcal{O})$; 2) In the M-step (Sec. 4.2), we learn the weights of the logic formulae in MLN, where $Q_\theta$ is fixed and $P_w$ is optimized to maximize the data log-likelihood.

## 4.1 E-STEP: INFERENCE

In the E-step, which is also known as the *inference* step, we are minimizing the KL divergence between the variational posterior distribution $Q_\theta(\mathcal{H}|\mathcal{O})$ and the true posterior distribution $P_w(\mathcal{H}|\mathcal{O})$. The exact inference of MLN is computationally intractable and proven to be NP-complete (Richardson & Domingos, 2006). Therefore, we choose to approximate the true posterior with a mean-field distribution, since the mean-field approximation has been demonstrated to scale up large graphical models, such as latent Dirichlet allocation for modeling topics from large text corpus (Hoffman et al., 2013). In the mean-field variational distribution, each unobserved ground predicate $r(a_r) \in \mathcal{H}$ is independently inferred as follows:

$$Q_\theta(\mathcal{H}|\mathcal{O}) := \prod_{r(a_r) \in \mathcal{H}} Q_\theta(r(a_r)), \quad (3)$$

where each factorized distribution $Q_\theta(r(a_r))$ follows the Bernoulli distribution. We parameterize the variational posterior $Q_\theta$ with deep learning models as our neural inference network. The design of the inference network is very important and has a lot of considerations, since we need a compact yet expressive model to accurately approximate the true posterior distribution. We employ graph neural networks with tunable embeddings as our inference network (detailed in Sec. 5), which can trade-off between the model compactness and expressiveness.

With the mean-field approximation, $\mathcal{L}_{\text{ELBO}}(Q_\theta, P_w)$ defined in Eq. 2 can be reorganized as below:

$$\Big(\sum_{f \in \mathcal{F}} w_f \sum_{a_f \in \mathcal{A}_f} \mathbb{E}_{Q_\theta(\mathcal{H}|\mathcal{O})}\Big[\phi_f(a_f)\Big] - \log Z(w)\Big) - \Big(\sum_{r(a_r) \in \mathcal{H}} \mathbb{E}_{Q_\theta(r(a_r))}\Big[\log Q_\theta(r(a_r))\Big]\Big), \quad (4)$$

where $w_f$ is fixed in the E-step and thus the partition function $Z(w)$ can be treated as a constant. We notice that the first term $\mathbb{E}_{Q_\theta(\mathcal{H}|\mathcal{O})}[\log P_w(\mathcal{O}, \mathcal{H})]$ has the summation over all formulae and all

possible assignments to each formula. Thus this double summation may involve a large number of terms. The second term $\mathbb{E}_{Q_\theta(\mathcal{H}|\mathcal{O})}[\log Q_\theta(\mathcal{H}|\mathcal{O})]$ is the sum of entropy of the variational posterior distributions $Q_\theta(r(a_r))$, which also involves a large number of terms since the summation ranges over all possible latent variables. Typically, the number of latent facts in database is much larger than the number of observed facts. Thus, both terms in the objective function pose the challenge of intractable computational cost.

To address this challenge, we sample mini-batches of ground formulae to break down the exponential summations by approximating it with a sequence of summations with a controllable number of terms. More specifically, in each optimization iteration, we first sample a batch of ground formulae. For each ground formula in the sampled batch, we compute the first term in Eq. 4 by taking the expectation of the corresponding potential function with respect to the posterior of the involved latent variables. The mean-field approximation enables us to decompose the global expectation over the entire MLN into local expectations over ground formulae. Similarly, for the second term in Eq. 4, we use the posterior of the latent variables in the sampled batch to compute a local sum of entropy.

For tasks that have sufficient labeled data as supervision, we can add a supervised learning objective to enhance the inference network, as follows:

$$\mathcal{L}_{\text{label}}(Q_\theta) = \sum_{r(a_r)\in\mathcal{O}} \log Q_\theta(r(a_r)). \tag{5}$$

This objective is complementary to the ELBO on predicates that are not well covered by logic rules but have sufficient observed facts. Therefore, the overall E-step objective function becomes:

$$\mathcal{L}_\theta = \mathcal{L}_{\text{ELBO}}(Q_\theta, P_w) + \lambda\mathcal{L}_{\text{label}}(Q_\theta), \tag{6}$$

where $\lambda$ is a hyperparameter to control the weight. This overall objective essentially combines the knowledge in logic rules and the supervision from labeled data.

## 4.2 M-STEP: LEARNING

In the M-step, which is also known as the *learning* step, we are learning the weights of logic formulae in Markov Logic Networks with the variational posterior $Q_\theta(\mathcal{H}|\mathcal{O})$ fixed. The partition function $Z(w)$ in Eq. 4 is not a constant anymore, since we need to optimize those weights in the M-step. There are exponential number of terms in the partition function $Z(w)$, which makes it intractable to directly optimize the ELBO. To tackle this problem, we adopt the widely used pseudo-log-likelihood (Richardson & Domingos, 2006) as an alternative objective for optimization, which is defined as:

$$P_w^*(\mathcal{O}, \mathcal{H}) := \mathbb{E}_{Q_\theta(\mathcal{H}|\mathcal{O})}\left[\sum_{r(a_r)\in\mathcal{H}} \log P_w(r(a_r) \mid \text{MB}_{r(a_r)})\right], \tag{7}$$

where $\text{MB}_{r(a_r)}$ is the Markov blanket of the ground predicate $r(a_r)$, i.e., the set of ground predicates that appear in some grounding of a formula with $r(a_r)$. For each formula $i$ that connects $r(a_r)$ to its Markov blanket, we optimize the formula weight $w_i$ by gradient descent, with the derivative:

$$\nabla_{w_i}\mathbb{E}_{Q_\theta}[\log P_w(r(a_r) \mid \text{MB}_{r(a_r)})] \simeq y_{r(a_r)} - P_w(r(a_r) \mid \text{MB}_{r(a_r)}), \tag{8}$$

where $y_{r(a_r)} = 0$ or 1 if $r(a_r)$ is an observed fact, and $y_{r(a_r)} = Q_\theta(r(a_r))$ otherwise. With the independence property of Markov Logic Networks, the gradients of the logic formulae weights can be efficiently computed on the Markov blanket of each variable.

For the M-step, we design a different sampling scheme to make it computationally efficient. For each variable in the Markov blanket, we take the truth value if it's observed and draw a sample from the variational posterior $Q_\theta$ if it's latent. In the M-step, the ELBO of a fully observed ground formula depends on the formula weight, thus we need to consider all the fully observed ground formulae. It is computationally intractable to use all possible ground predicates to compute the gradients in Eq. 8. To tackle this challenge, we simply consider all the ground formulae with at most one latent predicate, and pick up the ground predicate if its truth value determines the formula's truth value. Therefore, we keep a small subset of ground predicates, each of which can directly determine the truth value of a ground formula. Intuitively, this small subset contains all representative ground predicates, and makes good estimation of the gradients with much cheaper computational cost.

## 5 INFERENCE NETWORK DESIGN: EXPRESSGNN

In the neural variational EM framework, the key component is the posterior model, or the inference network. We need to design the inference network that is both expressive and efficient to approximate

the true posterior distribution. A recent concurrent work Qu & Tang (2019) uses a flattened embedding table as the entity representation to model the posterior. However, such simple posterior model is not able to capture the structure knowledge encoded in the knowledge graph. We employ graph neural networks with tunable embeddings to design our inference network. We also investigate the expressive power of GNN from theoretical perspective, which justifies our design.

Our inference network, named ExpressGNN, consists of three parts: the first part is a vanilla graph neural network (GNN), the second part uses tunable embeddings, and the third part uses the embeddings to define the variational posterior. For simplicity, we assume that each predicate has two arguments (i.e., consider only $r(c, c')$). We design each part as follows:

- We build a GNN on the knowledge graph $\mathcal{G}_{\mathcal{K}}$, which is much smaller than the ground graph of MLN (see comparison in Fig. 2). The computational graph of the GNN is given in Algorithm 1. The GNN parameters $\boldsymbol{\theta}_1$ and $\boldsymbol{\theta}_2$ are shared across the entire graph and independent of the number of entities. Therefore, the GNN is a compact model with $O(d^2)$ parameters given $d$ dimensional embeddings, $\mu_c \in \mathbb{R}^d$.

- For each entity in the knowledge graph, we augment its GNN embedding with a tunable embedding $\boldsymbol{\omega}_c \in \mathbb{R}^k$ as $\hat{\mu}_c = [\mu_c, \boldsymbol{\omega}_c]$. The tunable embeddings increase the expressiveness of the model. As there are $M$ entities, the number of parameters in tunable embeddings is $O(kM)$.

- We use the augmented embeddings of $c_1$ and $c_2$ to define the variational posterior. Specifically, $Q_\theta(r(c_1, c_2)) = \sigma(\texttt{MLP}_3(\hat{\mu}_{c_1}, \hat{\mu}_{c_2}, r; \boldsymbol{\theta}_3))$, where $\sigma(\cdot) = \frac{1}{1+\exp(-\cdot)}$. The number of parameters in $\boldsymbol{\theta}_3$ is $O(d + k)$.

In summary, ExpressGNN can be viewed as a two-level encoding of the entities: the compact GNN assigns similar embeddings to similar entities in the knowledge graph, while the expressive tunable embeddings provide additional model capacity to encode entity-specific information beyond graph structures. The overall number of trainable parameters in ExpressGNN is $O(d^2 + kM)$. By tuning the embedding size $d$ and $k$, ExpressGNN can trade-off between the model *compactness* and *expressiveness*. For large-scale problems with a large number of entities ($M$ is large), ExpressGNN can save a lot of parameters by reducing $k$.

---

**Algorithm 1:** GNN()

Initialize entity node: $\mu_c^{(0)} = \mu_0, \ \forall c \in \mathcal{C}$
**for** $t = 0$ *to* $T - 1$ **do**
    ▷ Compute message $\forall r(c, c') \in \mathcal{O}$
    $m_{c' \to c}^{(t)} = \texttt{MLP}_1(\mu_{c'}^{(t)}, r; \boldsymbol{\theta}_1)$
    ▷ Aggregate message $\forall c \in \mathcal{C}$
    $m_c^{(t+1)} = \texttt{AGG}(\{m_{c' \to c}^{(t)}\}_{c' : r(c, c') \in \mathcal{O}})$
    ▷ Update embedding $\forall c \in \mathcal{C}$
    $\mu_c^{(t+1)} = \texttt{MLP}_2(\mu_c^{(t)}, m_c^{(t+1)}; \boldsymbol{\theta}_2)$
**return** embeddings $\{\mu_c^{(T)}\}$

---

### 5.1 EXPRESSIVE POWER OF GNN AS INFERENCE NETWORK

The combination of GNN and tunable embeddings makes the model sufficiently expressive to approximate the true posterior distributions. Here we provide theoretical analysis on the expressive power of GNN in the mean-field inference problem, and discuss the benefit of combining GNN and tunable embeddings in ExpressGNN.

Recent studies (Shervashidze et al., 2011; Xu et al., 2018) show that the vanilla GNN embeddings can represent the results of graph coloring, but fail to represent the results of the more strict graph isomorphism check, i.e., GNN produces the same embedding for some nodes that should be distinguished. We first demonstrate this problem by a simple example:

**Example.** Fig. 3 involves four entities (A, B, E, F), two predicates (Friend: $\texttt{F}(\cdot, \cdot)$, Like: $\texttt{L}(\cdot, \cdot)$), and one formula ($\texttt{F}(c, c') \Rightarrow \texttt{L}(c, c')$). In this example, MLN variables have different posteriors, but GNN embeddings result in the same posterior representation. More specifically,

- Entity $A$ and $B$ have opposite relations with $E$, i.e., $\texttt{F}(A, E) = 1$ versus $\texttt{F}(B, E) = 0$ in the knowledge graph, but running GNN on the knowledge graph will always produce the same embeddings for $A$ and $B$, i.e., $\mu_A = \mu_B$.

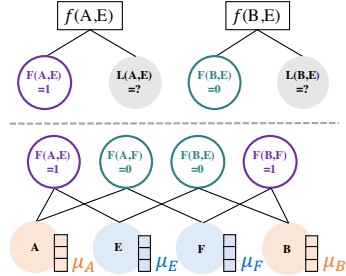

Figure 3: *Bottom*: A knowledge base with 0-1-0-1 loop. *Top*: MLN.

- $\mathtt{L}(A, E)$ and $\mathtt{L}(B, E)$ apparently have different posteriors.
  However, using GNN embeddings, $Q_\theta(\mathtt{L}(A, E)) = \sigma(\mathtt{MLP}_3(\mu_A, \mu_E, \mathtt{L}))$ is always identical to $Q_\theta(\mathtt{L}(B, E)) = \sigma(\mathtt{MLP}_3(\mu_B, \mu_E, \mathtt{L}))$.

We can formally prove that solving the problem in the above example requires the graph embeddings to distinguish any non-isomorphic nodes in the knowledge graph. A formal statement is provided below (see Appendix E for the proof).

**Definition 5.1.** Two ordered sequences of nodes $(c_1, \ldots, c_n)$ and $(c'_1, \ldots, c'_n)$ are *isomorphic* in a graph $\mathcal{G}_\mathcal{K}$ if there exists an isomorphism $\pi : \mathcal{G}_\mathcal{K} \to \mathcal{G}_\mathcal{K}$ such that $\pi(c_1) = c'_1, \ldots, \pi(c_n) = c'_n$.

**Theorem 5.1.** *Two latent variables $r(c_1, \ldots, c_n)$ and $r(c'_1, \ldots, c'_n)$ have the same posterior distribution in any MLN if and only if the nodes $(c_1, \cdots, c_n)$ and $(c'_1, \cdots, c'_n)$ are isomorphic in the knowledge graph $\mathcal{G}_\mathcal{K}$.*

Implied by the theorem, to obtain an expressive enough representation for the posterior, we need a more powerful GNN variant. A recent work has proposed a powerful GNN variant (Maron et al., 2019), which can handle small graphs such as chemical compounds and protein structures, but it is computationally expensive due to the usage of high-dimensional tensors. As a simple yet effective solution, ExpressGNN augments the vanilla GNN with additional tunable embeddings, which is a trade-off between the compactness and expressiveness of the model.

In summary, ExpressGNN has the following nice properties:

- *Efficiency*: ExpressGNN directly works on the knowledge graph, instead of the huge MLN grounding graph, making it much more efficient than the existing MLN inference methods.

- *Compactness*: The compact GNN model with shared parameters can be very memory efficient, making ExpressGNN possible to handle industry-scale problems.

- *Expressiveness*: The GNN model can capture structure knowledge encoded in the knowledge graph. Meanwhile, the tunable embeddings can encode entity-specific information, which compensates for GNN's deficiency in distinguishing non-isomorphic nodes.

- *Generalizability*: With the GNN embeddings, ExpressGNN may generalize to new entities or even different but related knowledge graphs unseen during training time without the need for retraining.

## 6  EXPERIMENTS

**Benchmark datasets.** We evaluate ExpressGNN and other baseline methods on four benchmark datasets: UW-CSE (Richardson & Domingos, 2006), Cora (Singla & Domingos, 2005), synthetic Kinship datasets, and FB15K-237 (Toutanova & Chen, 2015) constructed from Freebase (Bollacker et al., 2008). Details and full statistics of the benchmark datasets are provided in Appendix B.

**General settings.** We conduct all the experiments on a GPU-enabled (Nvidia RTX 2080 Ti) Linux machine powered by Intel Xeon Silver 4116 processors at 2.10GHz with 256GB RAM. We implement ExpressGNN using PyTorch and train it with Adam optimizer (Kingma & Ba, 2014). To ensure a fair comparison, we allocate the same computational resources (CPU, GPU and memory) for all the experiments. We use the default tuned hyperparameters for competitor methods, which can reproduce the experimental results reported in their original works.

**Model hyperparameters.** For ExpressGNN, we use 0.0005 as the initial learning rate, and decay it by half for every 10 epochs without improvement of validation loss. For Kinship, UW-CSE and Cora, we run ExpressGNN with a fixed number of iterations, and use the smallest subset from the original split for hyperparameter tuning. For FB15K-237, we use the original validation set to tune the hyperparameters. We use a two-layer MLP with ReLU activation function as the nonlinear transformation for each embedding update step in the GNN model. We learn different MLP parameters for different steps. To increase the model capacity of ExpressGNN, we also use different MLP parameters for different edge type, and for a different direction of embedding aggregation. For each dataset, we search the configuration of ExpressGNN on either the validation set or the smallest subset. The configuration we search includes the embedding size, the split point of tunable embeddings and GNN embeddings, the number of embedding update steps, and the sampling batch size. For the inference experiments, the weights for all the logic formulae are fixed as 1. For the

Table 1: Inference accuracy (AUC-PR) of different methods on three benchmark datasets.

| Method | Kinship | | | | | UW-CSE | | | | | Cora |
|---|---|---|---|---|---|---|---|---|---|---|---|
| | S1 | S2 | S3 | S4 | S5 | AI | Graphics | Language | Systems | Theory | (avg) |
| MCMC | 0.53 | - | - | - | - | - | - | - | - | - | - |
| BP / Lifted BP | 0.53 | 0.58 | 0.55 | 0.55 | 0.56 | 0.01 | 0.01 | 0.01 | 0.01 | 0.01 | - |
| MC-SAT | 0.54 | 0.60 | 0.55 | 0.55 | - | 0.03 | 0.05 | 0.06 | 0.02 | 0.02 | - |
| HL-MRF | **1.00** | **1.00** | **1.00** | **1.00** | - | 0.06 | 0.06 | 0.02 | 0.04 | 0.03 | - |
| ExpressGNN-E | 0.97 | 0.97 | 0.99 | 0.99 | 0.99 | **0.09** | **0.19** | **0.14** | **0.06** | **0.09** | **0.64** |

learning experiments, the weights are initialized as 1. For the choice of $\lambda$ in the combined objective $L_\theta$ in Eq. 6, we set $\lambda = 0$ for the inference experiments, since the query predicates are never seen in the training data and no supervision is available. For the learning experiments, we set $\lambda = 1$.

## 6.1 COMPARISON TO MLN INFERENCE METHODS AND ABLATION STUDY

We first evaluate the inference accuracy and efficiency of ExpressGNN. We compare our method with several strong MLN inference methods on UW-CSE, Cora and Kinship datasets. We also conduct ablation study to explore the trade-off between GNN and tunable embeddings.

**Experiment settings.** For the inference experiments, we fix the weights of all logic rules as 1. A key advantage of MLN is that it can handle open-world setting in a consistent probabilistic framework. Therefore, we adopt open-world setting for all the experiments, as opposed to closed-world setting where unobserved facts (except the query predicates) are assumed to be false. We also report the performance under closed-world setting in Appendix C.

**Prediction tasks.** The deductive logic inference task is to answer queries that typically involve single predicate. For example in UW-CSE, the task is to predict the AdvisedBy($c,c'$) relation for all persons in the set. In Cora, the task is to de-duplicate entities, and one of the query predicates is SameAuthor($c,c'$). As for Kinship, the task is to predict whether a person is male or female, i.e., Male($c$). For each possible substitution of the query predicate with different entities, the model is tasked to predict whether it's true or not.

**Evaluation metrics.** Following existing studies (Richardson & Domingos, 2006; Singla & Domingos, 2005), we use area under the precision-recall curve (AUC-PR) to evaluate the inference accuracy. To evaluate the inference efficiency, we use wall-clock running time in minutes.

**Competitor methods.** We compare our method with several strong MLN inference algorithms, including MCMC (Gibbs Sampling; Gilks et al. (1995); Richardson & Domingos (2006)), Belief Propagation (BP; Yedidia et al. (2001)), Lifted Belief Propagation (Lifted BP; Singla & Domingos (2008)), MC-SAT (Poon & Domingos, 2006) and Hinge-Loss Markov Random Field (HL-MRF; Bach et al. (2015); Srinivasan et al. (2019)).

**Inference accuracy.** The results of inference accuracy on three benchmark datasets are reported in Table 1. A hyphen in the entry indicates that it is either out of memory or exceeds the time limit (24 hours). We denote our method as ExpressGNN-E since only the E-step is needed for the inference experiments. Note that since the lifted BP is guaranteed to get identical results as BP (Singla & Domingos, 2008), the results of these two methods are merged into one row. For these experiments, ExpressGNN-E uses 64-dim GNN embeddings and 64-dim tunable embeddings. On Cora, all the baseline methods fail to handle the data scale under open-world setting, and ExpressGNN-E achieves good inference accuracy. On UW-CSE, ExpressGNN-E consistently outperforms all baselines. The Kinship dataset is synthesized and noise-free, and the number of entities increases linearly on the five sets S1–S5. HL-MRF achieves perfect accuracy for S1–S4, but is infeasible on the largest set S5. ExpressGNN-E yields similar but not perfect results, which is presumably caused by the stochastic nature of our sampling and optimization procedure.

**Inference efficiency.** The inference time corresponding to the experiments in Table 1 is summarized in Fig. 4. On UW-CSE (left table), ExpressGNN-E uses much shorter time for inference compared to all the baseline methods, and meanwhile ExpressGNN-E achieves the best inference performance. On Kinship (right figure), as the data size grows linearly from S1 to S5, the inference time of most baseline methods grows exponentially, while ExpressGNN-E maintains a nearly constant time cost, demonstrating its nice scalability. Some baseline methods such as MCMC and MC-SAT become

infeasible for larger sets. HL-MRF maintains a comparatively short inference time, however, it has a huge increase of memory cost and is not able to handle the largest set S5.

| Method | Inference Time (minutes) | | | | |
|---|---|---|---|---|---|
| | AI | Graphics | Language | Systems | Theory |
| MCMC | >24h | >24h | >24h | >24h | >24h |
| BP | 408 | 352 | 37 | 457 | 190 |
| Lifted BP | 321 | 270 | 32 | 525 | 243 |
| MC-SAT | 172 | 147 | 14 | 196 | 86 |
| HL-MRF | 135 | 132 | 18 | 178 | 72 |
| ExpressGNN-E | **14** | **20** | **5** | **7** | **13** |

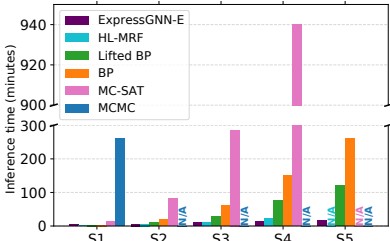

Figure 4: *Left / Right*: Inference time on UW-CSE / Kinship respectively. N/A indicates the method is infeasible.

**Ablation study.** ExpressGNN can trade-off the compactness and expressiveness of model by tuning the dimensionality of GNN and tunable embeddings. We perform ablation study on the Cora dataset to investigate how this trade-off affects the inference accuracy. Results of different configurations of ExpressGNN-E are shown in Table 2. It is observed that GNN64+Tune4 has comparable performance with Tune64, but is consistently better than GNN64. Note that the number of parameters in GNN64+Tune4 is $O(64^2 + 4|\mathcal{C}|)$, while that in Tune64 is $O(64|\mathcal{C}|)$. When the number of entities is large, GNN64+Tune4 has much less parameters to train. This is consistent with our theoretical analysis result: As a compact model, GNN saves a lot of parameters, but GNN alone is not expressive enough. A similar conclusion is observed

Table 2: AUC-PR for different combinations of GNN and tunable embeddings. Tune $d$ stands for $d$-dim tunable embeddings and GNN $d$ stands for $d$-dim GNN embeddings.

| Configuration | Cora | | | | |
|---|---|---|---|---|---|
| | S1 | S2 | S3 | S4 | S5 |
| Tune64 | 0.57 | 0.74 | 0.34 | 0.55 | 0.70 |
| GNN64 | 0.57 | 0.58 | 0.38 | 0.54 | 0.53 |
| GNN64+Tune4 | 0.61 | 0.75 | 0.39 | 0.54 | 0.70 |
| Tune128 | **0.62** | 0.76 | 0.42 | **0.60** | 0.73 |
| GNN128 | 0.60 | 0.59 | 0.45 | 0.55 | 0.61 |
| GNN64+Tune64 | **0.62** | **0.79** | **0.46** | 0.57 | **0.75** |

for GNN64+Tune64 and Tune128. Therefore, ExpressGNN seeks a combination of two types of embeddings to possess the advantage of both: having a compact model and being expressive. The best configuration of their embedding sizes can be varied on different tasks, and determined by the goal: getting a portable model or better performance.

## 6.2 COMPARISON TO KNOWLEDGE BASE COMPLETION METHODS

We evaluate ExpressGNN in the knowledge base completion task on the FB15K-237 dataset, and compare it with state-of-the-art knowledge base completion methods.

**Experiment settings.** To generate logic rules, we use Neural LP (Yang et al., 2017) on the training set and pick up the candidates with top confidence scores. See Appendix D for examples of selected logic rules. We evaluate both inference-only and inference-and-learning version of ExpressGNN, denoted as ExpressGNN-E and ExpressGNN-EM, respectively.

**Prediction task.** For each test query $r(c, c')$ with respect to relation $r$, the model is tasked to generate a rank list over all possible instantiations of $r$ and sort them according to the model's confidence on how likely this instantiation is true.

**Evaluation metrics.** Following existing studies (Bordes et al., 2013; Sun et al., 2019), we use filtered ranking where the test triples are ranked against all the candidate triples not appearing in the dataset. Candidate triples are generated by corrupting the subject or object of a query $r(c, c')$. For evaluation, we compute the Mean Reciprocal Ranks (MRR), which is the average of the reciprocal rank of all the truth queries, and Hits@10, which is the percentage of truth queries that are ranked among the top 10.

**Competitor methods.** Since none of the aforementioned MLN inference methods can scale up to this dataset, we compare ExpressGNN with a number of state-of-the-art methods for knowledge base completion, including Neural Tensor Network (NTN; Socher et al. (2013)), Neural LP (Yang et al., 2017), DistMult (Kadlec et al., 2017), ComplEx (Trouillon et al., 2016), TransE (Bordes et al., 2013), RotatE (Sun et al., 2019) and pLogicNet (Qu & Tang, 2019). The results of MLN and pLogicNet are directly taken from the paper Qu & Tang (2019). For all the other baseline methods, we use publicly available code with the provided best hyperparameters to run the experiments.

Table 3: Performance on FB15K-237 with varied training set size.

| Model | MRR | | | | | Hits@10 | | | | |
|---|---|---|---|---|---|---|---|---|---|---|
| | 0% | 5% | 10% | 20% | 100% | 0% | 5% | 10% | 20% | 100% |
| MLN | - | - | - | - | 0.10 | - | - | - | - | 16.0 |
| NTN | 0.09 | 0.10 | 0.10 | 0.11 | 0.13 | 17.9 | 19.3 | 19.1 | 19.6 | 23.9 |
| Neural LP | 0.01 | 0.13 | 0.15 | 0.16 | 0.24 | 1.5 | 23.2 | 24.7 | 26.4 | 36.2 |
| DistMult | 0.23 | 0.24 | 0.24 | 0.24 | 0.31 | 40.0 | 40.4 | 40.7 | 41.4 | 48.5 |
| ComplEx | 0.24 | 0.24 | 0.24 | 0.25 | 0.32 | 41.1 | 41.3 | 41.9 | 42.5 | 51.1 |
| TransE | 0.24 | 0.25 | 0.25 | 0.25 | 0.33 | 42.7 | 43.1 | 43.4 | 43.9 | 52.7 |
| RotatE | 0.25 | 0.25 | 0.25 | 0.26 | 0.34 | 42.6 | 43.0 | 43.5 | 44.1 | 53.1 |
| pLogicNet | - | - | - | - | 0.33 | - | - | - | - | 52.8 |
| ExpressGNN-E | **0.42** | **0.42** | 0.42 | 0.44 | 0.45 | 53.1 | 53.1 | 53.3 | 55.2 | 57.3 |
| ExpressGNN-EM | **0.42** | **0.42** | **0.43** | **0.45** | **0.49** | **53.8** | **54.6** | **55.3** | **55.6** | **60.8** |

Table 4: Zero-shot learning performance on FB15K-237.

| Model | MRR | Hits@10 |
|---|---|---|
| NTN | 0.001 | 0.0 |
| Neural LP | 0.010 | 2.7 |
| DistMult | 0.004 | 0.8 |
| ComplEx | 0.013 | 2.2 |
| TransE | 0.003 | 0.5 |
| RotatE | 0.006 | 1.5 |
| ExpressGNN-E | 0.181 | 29.3 |
| ExpressGNN-EM | **0.185** | **29.6** |

**Performance analysis.** The experimental results on the full training data are reported in Table 3 (100% columns). Both ExpressGNN-E and ExpressGNN-EM significantly outperform all the baseline methods. With learning the weights of logic rules, ExpressGNN-EM achieves the best performance. Compared to MLN, ExpressGNN achieves much better performance since MLN only relies on the logic rules while ExpressGNN can also leverage the labeled data as additional supervision. Compared to knowledge graph embedding methods such as TransE and RotatE, ExpressGNN can leverage the prior knowledge in logic rules and outperform these purely data-driven methods.

**Data efficiency.** We investigate the data efficiency of ExpressGNN and compare it with baseline methods. Following (Yang et al., 2017), we split the knowledge base into facts / training / validation / testing sets, and vary the size of the training set from 0% to 100% to feed the model with complete facts set for training. From Table 3, we see that ExpressGNN performs significantly better than the baselines on smaller training data. With more training data as supervision, data-driven baseline methods start to close the gap with ExpressGNN. This clearly shows the benefit of leveraging the knowledge encoded in logic rules when there data is insufficient for supervised learning.

**Zero-shot relational learning.** In practical scenarios, a large portion of the relations in the knowledge base are long-tail, i.e., most relations may have only a few facts (Xiong et al., 2018). Therefore, it is important to investigate the model performance on relations with insufficient training data. We construct a zero-shot learning dataset based on FB15K-237 by forcing the training and testing data to have disjoint sets of relations. Table 4 shows the results. As expected, the performance of all the supervised relational learning methods drop to almost zero. This shows the limitation of such methods when coping with sparse long-tail relations. Neural LP is designed to handle new entities in the test set (Yang et al., 2017), but still struggles to perform well in zero-shot learning. In contrast, ExpressGNN leverages both the prior knowledge in logic rules and the neural relational embeddings for reasoning, which is much less affected by the scarcity of data on long-tail relations. Both variants of our framework (ExpressGNN-E and ExpressGNN-EM) achieve significantly better performance.

# 7 CONCLUSION

This paper studies the probabilistic logic reasoning problem, and proposes ExpressGNN to combine the advantages of Markov Logic Networks in logic reasoning and graph neural networks in graph representation learning. ExpressGNN addresses the scalability issue of Markov Logic Networks with efficient stochastic training in the variational EM framework. ExpressGNN employs GNNs to capture the structure knowledge that is implicitly encoded in the knowledge graph, which serves as supplement to the knowledge from logic formulae. ExpressGNN is a general framework that can trade-off the model compactness and expressiveness by tuning the dimensionality of the GNN and the embedding part.

ACKNOWLEDGEMENTS

We acknowledge grants from NSF IIS-1218749, NIH BIGDATA 1R01GM108341, NSF CAREER IIS-1350983, NSF IIS-1639792 EAGER, NSF IIS-1841351 EA-GER, NSF CNS-1704701, ONR N00014-15-1-2340, Intel ISTC, Nvidia, Google, Amazon AWS and Siemens. We thank Hyunsu Park for his insightful discussions and help on experiments. We thank the anonymous reviewers for their helpful and thoughtful comments. Yuyu Zhang is supported by the Siemens FutureMaker Fellowship.

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

# Appendix

## A  COUNTER EXAMPLES

We provide more examples in this section to show that it is more than a rare case that GNN embeddings alone are not expressive enough.

### A.1  EXAMPLE 1

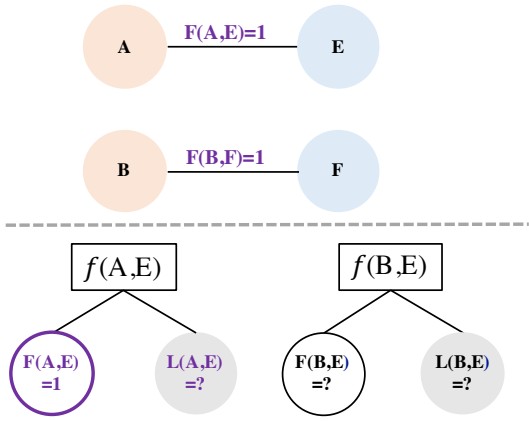

Figure 5: Example 1. *Top*: Knowledge base. *Bottom*: MLN

Unlike the example shown in main text, where A and B have OPPOSITE relation with E, Fig. 5 shows a very simple example where A and B have exactly the same structure which makes A and B indistinguishable and isomorphic. However, since (A,E) and (B,E) are not isomorphic, it can be easily seen that L(A, E) has different posterior from L(B, E).

### A.2  EXAMPLE 2

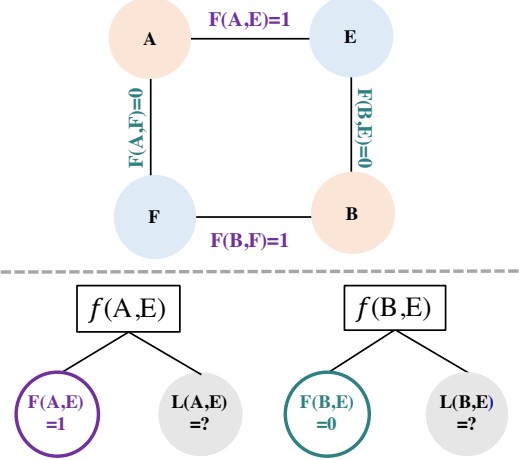

Figure 6: The same example as in Fig. 3. *Top*: Knowledge base. *Bottom*: MLN

Fig. 6 shows an example which is the same as in Fig. 3. However, in this example, it is already revealed in the knowledge base that $(A, E)$ and $(B, E)$ have different local structures as they are connected by different observations. That is, $(A, [F(A, E) = 1], E)$ and $(B, [F(B, E) = 0], E)$ can be distinguished by GNN.

Now, we use another example in Fig. 7 to show that even when the local structures are the same, the posteriors can still be different, which is caused by the formulae.

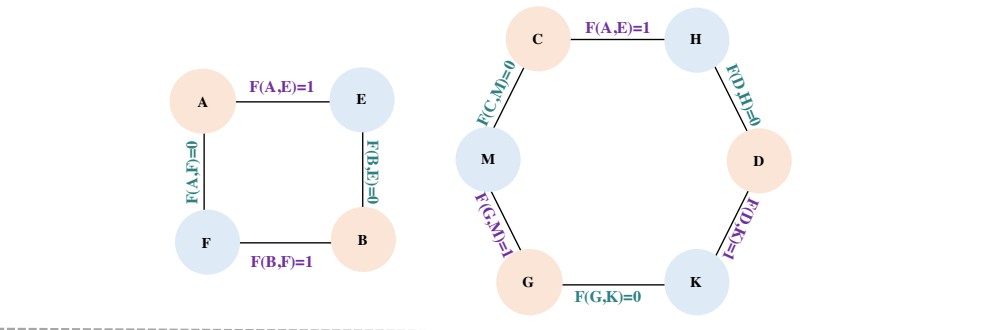

$$f(c_1, c_2, c_3, c_4) := \big(F(c_1, c_2) \wedge \neg F(c_3, c_2) \wedge F(c_3, c_4) \wedge \neg F(c_1, c_4)\big) \Rightarrow L(c_1, c_2)$$

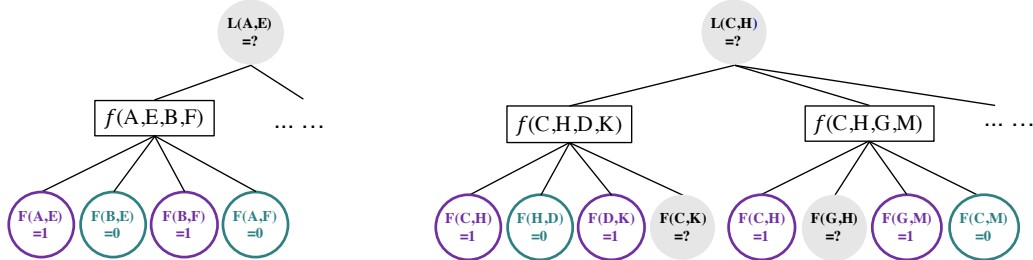

Figure 7: Example 2. *Top*: Knowledge base. *Bottom*: MLN

In Fig. 7, $(A, E)$ and $(C, H)$ have the same local structure, so that the tuple $(A, [F(A, E) = 1], E)$ and $(C, [F(C, H) = 1], H)$ can NOT be distingushed by GNN. However, we can make use of subgraph $(A, E, B, F)$ to define a formula, and then the resulting MLN gives different posterior to $L(A, E)$ and $L(C, H)$, as can be seen from the figure. Note that this construction of MLN is the same as the construction steps stated in the proof in Sec. E.

Table 5: Complete statistics of the benchmark datasets.

| Dataset | # entity | # relation | # fact | # query | # ground predicate | # ground formula |
|---|---|---|---|---|---|---|
| FB15K-237 | 15K | 237 | 272K | 20K | 50M | 679B |
| Kinship-S1 | 62 | 15 | 187 | 38 | 50K | 550K |
| Kinship-S2 | 110 | 15 | 307 | 62 | 158K | 3M |
| Kinship-S3 | 160 | 15 | 482 | 102 | 333K | 9M |
| Kinship-S4 | 221 | 15 | 723 | 150 | 635K | 23M |
| Kinship-S5 | 266 | 15 | 885 | 183 | 920K | 39M |
| UW-CSE-AI | 300 | 22 | 731 | 4K | 95K | 73M |
| UW-CSE-Graphics | 195 | 22 | 449 | 4K | 70K | 64M |
| UW-CSE-Language | 82 | 22 | 182 | 1K | 15K | 9M |
| UW-CSE-Systems | 277 | 22 | 733 | 5K | 95K | 121M |
| UW-CSE-Theory | 174 | 22 | 465 | 2K | 51K | 54M |
| Cora-S1 | 670 | 10 | 11K | 2K | 175K | 621B |
| Cora-S2 | 602 | 10 | 9K | 2K | 156K | 431B |
| Cora-S3 | 607 | 10 | 18K | 3K | 156K | 438B |
| Cora-S4 | 600 | 10 | 12K | 2K | 160K | 435B |
| Cora-S5 | 600 | 10 | 11K | 2K | 140K | 339B |

## B  DATASET DETAILS

For our experiments, we use the following benchmark datasets:

- The social network dataset UW-CSE (Richardson & Domingos, 2006) contains publicly available information of students and professors in the CSE department of UW. The dataset is split into five sets according to the home department of the entities.
- The entity resolution dataset Cora (Singla & Domingos, 2005) consists of a collection of citations to computer science research papers. The dataset is also split into five subsets according to the field of research.
- We introduce a synthetic dataset that resembles the popular Kinship dataset (Denham, 1973). The original dataset contains kinship relationships (e.g., Father, Brother) among family members in the Alyawarra tribe from Central Australia. The synthetic dataset closely resembles the original Kinship dataset but with a controllable number of entities. To generate a dataset with $n$ entities, we randomly split $n$ entities into two groups which represent the first and second generation respectively. Within each group, entities are grouped into a few sub-groups representing the sister- and brother-hood. Finally, entities from different sub-groups in the first generation are randomly coupled and a sub-group in the second generation is assigned to them as their children. To generate the knowledge base, we traverse this family tree, and record all kinship relations for each entity. We generate five kinship datasets (Kinship S1–S5) by linearly increasing the number of entities.
- The knowledge base completion benchmark FB15K-237 (Toutanova & Chen, 2015) is a generic knowledge base constructed from Freebase (Bollacker et al., 2008), which is designed to a more challenging variant of FB15K. More specifically, FB15K-237 is constructed by removing near-duplicate and inverse relations from FB15K. The dataset is split into training / validation / testing and we use the same split of facts from training as in prior work (Yang et al., 2017).

The complete statistics of these datasets are shown in Table 5. Examples of logic formulae used in four benchmark datasets are listed in Table 7.

## C  INFERENCE WITH CLOSED-WORLD SEMANTICS FOR BASELINE METHODS

In Sec. 6.1 we compare ExpressGNN with five probabilistic inference methods under open-world semantics. This is different from the original works, where they generally adopt the closed-world setting due to the scalability issues. More specifically, the original works assume that the predicates (except the ones in the query) observed in the knowledge base is *closed*, meaning for all instantiations

Table 6: Inference performance of competitors and our method under the closed-world semantics.

| Method | Cora | | | | | UW-CSE | | | | |
|---|---|---|---|---|---|---|---|---|---|---|
| | S1 | S2 | S3 | S4 | S5 | AI | Graphics | Language | Systems | Theory |
| MCMC | 0.43 | 0.63 | 0.24 | 0.46 | 0.56 | 0.19 | 0.04 | 0.03 | 0.15 | 0.08 |
| BP / Lifted BP | 0.44 | 0.62 | 0.24 | 0.45 | 0.57 | 0.21 | 0.04 | 0.01 | 0.14 | 0.05 |
| MC-SAT | 0.43 | 0.63 | 0.24 | 0.46 | 0.57 | 0.13 | 0.04 | 0.03 | 0.11 | 0.08 |
| HL-MRF | 0.60 | 0.78 | 0.52 | 0.70 | 0.81 | 0.26 | 0.18 | 0.06 | 0.27 | 0.19 |

of these predicates that do not appear in the knowledge base are considered *false*. Note that only the query predicates remain open-world in this setting.

For sanity checking, we also conduct these experiments with a closed-world setting. We found the results summarized in Table 6 are close to those reported in the original works. This shows that we have a fair setup (including memory size, hyperparameters, etc.) for those competitor methods. Additionally, one can find that the AUC-PR scores compared to those (Table 1) under open-world setting are actually better. This is due to the way the datasets were originally collected and evaluated generally complies with the closed-world assumption. But this is very unlikely to be true for real-world and large-scale knowledge base such as Freebase and WordNet, where many *true* facts between entities are not observed. Therefore, in general, the open-world setting is much more reasonable, which we follow throughout this paper.

## D    LOGIC FORMULAE

We list some examples of logic formulae used in four benchmark datasets in Table 7. The full list of logic formulae is available in our source code repository. Note that these formulae are not necessarily as clean as being always true, but are typically true.

For UW-CSE and Cora, we use the logic formulae provided in the original dataset. UW-CSE provides 94 hand-coded logic formulae, and Cora provides 46 hand-coded rules. For Kinship, we hand-code 22 first-order logic formulae. For FB15K-237, we first use Neural LP (Yang et al., 2017) on the full data to generate candidate rules. Then we select the ones that have confidence scores higher than 90% of the highest scored formulae sharing the same target predicate. We also de-duplicate redundant rules that can be reduced to other rules by switching the logic variables. Finally, we have generated 509 logic formulae for FB15K-237.

Table 7: Examples of logic formulae used in four benchmark datasets.

| Dataset | First-order Logic Formulae |
|---------|---------------------------|
| Kinship | `Father(X,Z)` $\wedge$ `Mother(Y,Z)` $\Rightarrow$ `Husband(X,Y)`
`Father(X,Z)` $\wedge$ `Husband(X,Y)` $\Rightarrow$ `Mother(Y,Z)`
`Husband(X,Y)` $\Rightarrow$ `Wife(Y,X)`
`Son(Y,X)` $\Rightarrow$ `Father(X,Y)` $\vee$ `Mother(X,Y)`
`Daughter(Y,X)` $\Rightarrow$ `Father(X,Y)` $\vee$ `Mother(X,Y)` |
| UW-CSE | `taughtBy(c, p, q)` $\wedge$ `courseLevel(c, Level500)` $\Rightarrow$ `professor(p)`
`tempAdvisedBy(p, s)` $\Rightarrow$ `professor(p)`
`advisedBy(p, s)` $\Rightarrow$ `student(s)`
`tempAdvisedBy(p, s)` $\Rightarrow$ `student(s)`
`professor(p)` $\wedge$ `hasPosition(p, Faculty)` $\Rightarrow$ `taughtBy(c, p, q)` |
| Cora | `SameBib(b1,b2)` $\wedge$ `SameBib(b2,b3)` $\Rightarrow$ `SameBib(b1,b3)`
`SameTitle(t1,t2)` $\wedge$ `SameTitle(t2,t3)` $\Rightarrow$ `SameTitle(t1,t3)`
`Author(bc1,a1)` $\wedge$ `Author(bc2,a2)` $\wedge$ `SameAuthor(a1,a2)` $\Rightarrow$ `SameBib(bc1,bc2)`
`HasWordVenue(a1, +w)` $\wedge$ `HasWordVenue(a2, +w)` $\Rightarrow$ `SameVenue(a1, a2)`
`Title(bc1,t1)` $\wedge$ `Title(bc2,t2)` $\wedge$ `SameTitle(t1,t2)` $\Rightarrow$ `SameBib(bc1,bc2)` |
| FB15K-237 | `position(B, A)` $\wedge$ `position(C, B)` $\Rightarrow$ `position(C, A)`
`ceremony(B, A)` $\wedge$ `ceremony(C, B)` $\Rightarrow$ `categoryOf(C, A)`
`film(B, A)` $\wedge$ `film(C, B)` $\Rightarrow$ `participant(A, C)`
`storyBy(A, B)` $\Rightarrow$ `participant(A, B)`
`adjoins(A, B)` $\wedge$ `country(B, C)` $\Rightarrow$ `serviceLocation(A, C)` |

# E    PROOF OF THEOREM

First, we re-state the definition and theorem in a more mathematical form:

**Definition.** [Isomorphic Nodes] Two ordered sequences of nodes $(c_1, \ldots, c_n)$ and $(c'_1, \ldots, c'_n)$ are **isomorphic** in a graph $\mathcal{G}_\mathcal{K}$ if there exists an isomorphism from $\mathcal{G}_\mathcal{K} = (\mathcal{C}, \mathcal{O}, \mathcal{E})$ to itself, i.e., $\pi : \mathcal{C} \cup \mathcal{O} \to \mathcal{C} \cup \mathcal{O}$, such that $\pi(c_1) = c'_1, \ldots, \pi(c_n) = c'_n$. Further, we use the following notation

$$(c_1, \cdots, c_n) \overset{\mathcal{G}_\mathcal{K}}{\Longleftrightarrow} (c'_1, \cdots, c'_n) : (c_1, \cdots, c_n) \text{ and } (c'_1, \cdots, c'_n) \text{ are isomorphic in } \mathcal{G}_\mathcal{K}.$$

**Theorem.** Consider a knowledge base $\mathcal{K} = (\mathcal{C}, \mathcal{R}, \mathcal{O})$ and any $r \in \mathcal{R}$. Two latent random variables $X := r(c_1, \ldots, c_n)$ and $X' := r(c'_1, \ldots, c'_n)$ have the same posterior distribution in **any** MLN **if and only if** $(c_1, \cdots, c_n) \overset{\mathcal{G}_\mathcal{K}}{\Longleftrightarrow} (c'_1, \cdots, c'_n)$.

Then we give the proof as follows.

*Proof.* A graph isomorphism from $G$ to itself is called automorphism, so in this proof, we will use the terminology - automorphism - to indicate such a self-bijection.

($\Longleftarrow$) We first prove the sufficient condition:

If $\exists$ automorphism $\pi$ on the graph $\mathcal{G}_\mathcal{K}$ such that $\pi(c_i) = c'_i, \forall i = 1, ..., n,$

then for any $r \in \mathcal{R}$, $r(c_1, \ldots, c_n)$ and $r(c'_1, \ldots, c'_n)$ have the same posterior in any MLN.

MLN is a graphical model that can also be represented by a factor graph $\text{MLN} = (\mathcal{O} \cup \mathcal{H}, \mathcal{F}_g, \mathcal{E})$ where ground predicates (random variables) and ground formulae (potential) are connected. We will show that $\exists$ an automorphism $\phi$ on MLN such that $\phi(r(c_1, \ldots, c_n)) = r(c'_1, \ldots, c'_n)$. Then the sufficient condition is true. This automorphism $\phi$ is easy to construct using the automorphism $\pi$ on $\mathcal{G}_\mathcal{K}$. More precisely, we define $\phi : (\mathcal{O} \cup \mathcal{H}, \mathcal{F}_g) \to (\mathcal{O} \cup \mathcal{H}, \mathcal{F}_g)$ as

$$\phi(r(a_r)) = r(\pi(a_r)), \quad \phi(f(a_f)) = f(\pi(a_f)), \tag{9}$$

for any predicate $r \in \mathcal{R}$, any assignments $a_r$ to its arguments, any formula $f \in \mathcal{F}$, and any assignments $a_f$ to its arguments. It is easy to see $\phi$ is an automorphism:

1. Since $\pi$ is a bijection, apparently $\phi$ is also a bijection.
2. The above definition preserves the biding of the arguments. $r(a_r)$ and $f(a_f)$ are connected if and only if $\phi(r(a_r))$ and $f(\pi(a_f))$ are connected.
3. Given the definition of $\pi$, we know that $r(a_r)$ and $r(\pi(a_r))$ have the same observation value. Therefore, in MLN, $\texttt{NodeType}(r(a_r)) = \texttt{NodeType}(\phi(r(a_r)))$.

This completes the proof of the sufficient condition.

($\Longrightarrow$) To prove the necessary condition, it is equivalent to show the following assumption

**(A 1)**: there is no automorphism $\pi$ on the graph $\mathcal{G}_\mathcal{K}$ such that $\pi(c_i) = c'_i, \forall i = 1, ..., n,$

can imply:

there must exists a MLN and a predicate $r$ in it such that

$r(c_1, \ldots, c_n)$ and $r(c'_1, \ldots, c'_n)$ have different posterior.

Before showing this, let us first introduce the **factor graph representation of a single logic formula** $f$.

A logic formula $f$ can be represented as a factor graph, $\mathcal{G}_f = (\mathcal{C}_f, \mathcal{R}_f, \mathcal{E}_f)$, where nodes on one side of the graph is the set of distinct constants $\mathcal{C}_f$ needed in the formula, while nodes on the other side is the set of predicates $\mathcal{R}_f$ used to define the formula. The set of edges, $\mathcal{E}_f$, will connect constants to predicates or predicate negation. That is, an edge

$$e = (c, r, i) \text{ between}$$
node $c$ and predicate $r$ exists, if the predicate $r$ use constant $c$ in its $i$-th argument.

We note that the set of distinctive constants used in the definition of logic formula are templates where actual constant can be instantiated from $\mathcal{C}$. An illustration of logic formula factor graph can be found in Fig. 8. Similar to the factor graph for the knowledge base, we also differentiate the type of edges by the position of the argument.

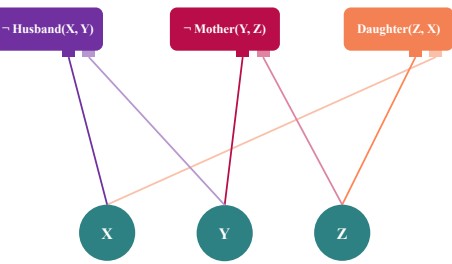

Figure 8: An example of the factor graph for the logic formula ¬Husband(X,Y) ∨ ¬Mother(Y,Z) ∨ Daughter(Z,X).

Therefore, every single formula can be represented by a factor graph. We will construct a factor graph representation to define a particular formula, and show that the MLN induced by this formula will result in different posteriors for $r(c_1, \ldots, c_n)$ and $r(c'_1, \ldots, c'_n)$. The factor graph for the formula is constructed in the following way (see Fig. 7 as an example of the resulting formula constructed using the following steps):

(i) Given the above assumption **(A 1)**, we claim that:

$\exists$ a subgraph $\mathcal{G}^*_{c_{1:n}} = (\mathcal{C}^*_c, \mathcal{O}^*_c, \mathcal{E}^*_c) \subseteq \mathcal{G}_{\mathcal{K}}$ such that all subgraphs $\mathcal{G}_{c'_{1:n}} = (\mathcal{C}_{c'}, \mathcal{O}_{c'}, \mathcal{E}_{c'}) \subseteq \mathcal{G}_{\mathcal{K}}$ satisfy:

**(Condition)** if there exists an isomorphism $\phi : \mathcal{G}^*_{c_{1:n}} \rightarrow \mathcal{G}_{c'_{1:n}}$ satisfying $\phi(c_i) = c'_i, \forall i = 1, \ldots, n$ after the observation values are IGNORED (that is, $[r_j(\cdots) = 0]$ and $[r_j(\cdots) = 1]$ are treated as the SAME type of nodes), then the set of fact nodes (observations) in these two graphs are different (that is, $\mathcal{O}^*_c \neq \mathcal{O}_{c'}$).

The proof of this claim is given at the end of this proof.

(ii) Next, we use $\mathcal{G}^*_{c_{1:n}}$ to define a formula $f$. We first initialize the definition of the formula value as

$$f(c_1, \ldots, c_n, \tilde{c}_1, \ldots, \tilde{c}_n) = \left( \wedge \left\{ \tilde{r}(a_{\tilde{r}}) : \tilde{r}(a_{\tilde{r}}) \in \mathcal{G}^*_{c_{1:n}} \right\} \right) \Rightarrow r(c_1, \ldots, c_n). \quad (10)$$

Then, we change $\tilde{r}(a_{\tilde{r}})$ in this formula to the negation $\neg \tilde{r}(a_{\tilde{r}})$ if the observed value of $\tilde{r}(a_{\tilde{r}})$ is 0 in $\mathcal{G}^*_{c_{1:n}}$.

We have defined a formula $f$ using the above two steps. Suppose the MLN only contains this formula $f$. Then

the two nodes $r(c_1, \ldots, c_n)$ and $r(c'_1, \ldots, c'_n)$ in this MLN must be distinguishable.

The reason is, in MLN, $r(c_1, \ldots, c_n)$ is connected to a ground formula $f(c_1, \ldots, c_n, \tilde{c}_1, \ldots, \tilde{c}_n)$, whose factor graph representation is $\mathcal{G}^*_{c_{1:n}} \cup r(c_1, \ldots, c_n)$. In this formula, all variables are observed in the knowledge base $\mathcal{K}$ except for $r(c_1, \ldots, c_n)$ and and the observation set is $\mathcal{O}^*_c$. The formula value is

$$f(c_1, \ldots, c_n, \tilde{c}_1, \ldots, \tilde{c}_n) = (1 \Rightarrow r(c_1, \ldots, c_n)). \quad (11)$$

Clarification: Eq. 10 is used to **define** a formula and $c_i$ in this equation can be replaced by other constants, while Eq. 11 represents a **ground** formula whose arguments are exactly $c_1, \ldots, c_n, \tilde{c}_1, \ldots, \tilde{c}_n$. Based on **(Condition)**, there is NO formula $f(c'_1, \ldots, c'_n, \tilde{c}'_1, \ldots, \tilde{c}'_n)$ that contains $r(c'_1, \ldots, c'_n)$ has

an observation set the same as $\mathcal{O}_c^*$. Therefore, $r(c_1, \ldots, c_n)$ and $r(c'_1, \ldots, c'_n)$ are distinguishable in this MLN.

Proof of claim:

We show the existence by constructing the subgraph $\mathcal{G}_{c_{1:n}}^* \subseteq \mathcal{G}_{\mathcal{K}}$ in the following way:

(i) First, we initialize the subgraph as $\mathcal{G}_{c_{1:n}}^* := \mathcal{G}_{\mathcal{K}}$. Given assumption **(A 1)** stated above, it is clear that

> **(S 1)** $\forall$ subgraph $\mathcal{G}' \subseteq \mathcal{G}_{\mathcal{K}}$, there is no isomorphism $\pi : \mathcal{G}_{c_{1:n}}^* \to \mathcal{G}'$ satisfying $\pi(c_i) = c'_i, \forall i = 1, \ldots, n$.

(ii) Second, we need to check wether the following case occurs:

> **(C 1)** $\exists$ a subgraph $\mathcal{G}' = (\mathcal{C}', \mathcal{O}', \mathcal{E}')$ such that (1) there EXISTS an isomorphism $\phi : \mathcal{G}_{c_{1:n}}^* \to \mathcal{G}'$ satisfying $\phi(c_i) = c'_i, \forall i = 1, \ldots, n$ after the observation values are IGNORED (that is, $[r_j(\cdots) = 0]$ and $[r_j(\cdots) = 1]$ are treated as the same type of nodes); and (2) the set of factor nodes (observations) in these two graphs are the same (that is, $\mathcal{O}_c^* = \mathcal{O}'$).

(iii) Third, we need to modify the subgraph if the case **(C 1)** is observed. Since $\left|\mathcal{G}_{c_{1:n}}^*\right| \geq |\mathcal{G}'|$, the only subgraph that will lead to the case **(C1)** is the maximal subgraph $\mathcal{G}_{c_{1:n}}^*$. The isomorphism $\phi$ is defined by ignoring the observation values, while the isomorphism $\pi$ in **(S 1)** is not ignoring them. Thus,

> **(S 1)** and **(C 1)** $\implies$ $\exists$ a set of nodes $S := \left\{ \left[r_j(a^{(1)}) = 0\right], \ldots, \left[r_j(a^{(n)}) = 0\right] \right\}$ such that for any isomorphism $\phi$ satisfying the conditions in **(C 1)**, the range $\phi(S)$ contains at least one node $[r_j(\cdot) = 1]$ which has observation value 1.

Otherwise, it is easy to see a contradiction to statement **(S 1)**.

> **(M 1)** Modify the subgraph by $\mathcal{G}_{c_{1:n}}^* \longleftarrow \mathcal{G}_{c_{1:n}}^* \setminus S$. The nodes (and also their edges) in the set $S := \left\{ \left[r_j(a^{(1)}) = 0\right], \ldots, \left[r_j(a^{(n)}) = 0\right] \right\}$ are removed.

For the new subgraph $\mathcal{G}_{c_{1:n}}^*$ after the modification **(M 1)**, the case **(C 1)** will not occur. Thus, we've obtained a subgraph that satisfies the conditions stated in the claim. Finally, we can remove the nodes that are not connected with $\{c_1, \ldots, c_n\}$ (that is, there is no path between this node and any one of $\{c_1, \ldots, c_n\}$). The remaining graph is connected to $\{c_1, \ldots, c_n\}$ and still satisfies the conditions that we need.

$\square$

