# OpenReview forum: "Efficient Probabilistic Logic Reasoning with Graph Neural Networks"
_ICLR.cc/2020/Conference — Accept (Poster)_

### Official Review · AnonReviewer2 · 2019-10-22
**Official Blind Review #2**

**Rating:** 3

**Review:**

In this paper the authors propose a system, called ExpressGNN, that combines MLNs and GNNs. This system is able to perform inference and learning the weights of the logic formulas.

The proposed approach seems valid and really intriguing. Moreover the problems it tackles, i.e. inference and learning over big knowledge graphs, are of foremost importance and are interesting for a wide community of researchers.
I have just one concern and it is about the experiments for the knowledge graph completion task. In fact, this task was performed only on one KG. I think the proposed system should be evaluated on more KGs.

For these reasons I think the paper, after an extension of the experimental results, should be accepted.

[Minor]
Page 3. “The equality holds” which equality are you talking about?


**Experience Assessment:**

I have read many papers in this area.

**Review Assessment: Checking Correctness Of Derivations And Theory:**

I did not assess the derivations or theory.

**Review Assessment: Checking Correctness Of Experiments:**

I assessed the sensibility of the experiments.

**Review Assessment: Thoroughness In Paper Reading:**

I made a quick assessment of this paper.

---

> ### Author Response · Authors · 2019-11-14
> **Response to Reviewer #2**
>
> Thanks for your review comments. We briefly respond to your questions as follows.
>
>
> > The proposed system should be evaluated on more KGs.
>
> In fact, our method is evaluated on four benchmark datasets with four different KGs: UW-CSE, Cora, Kinship, and Freebase. These knowledge graphs are of different knowledge types and data distributions, and are widely used as benchmark datasets to evaluate MLNs and knowledge graph reasoning methods.
>
>
> > Page 3. “The equality holds” which equality are you talking about?
>
> “The equality holds” points to the equality in Eq. (2). To make it more clear, we have added a reference to Eq. (2) in the updated paper.

---

### Official Review · AnonReviewer1 · 2019-10-23
**Official Blind Review #1**

**Rating:** 3

**Review:**

This paper proposes a framework for solving the probabilistic logic reasoning problem by integrating Markov neural networks and graph neural networks to combine their individual features into a more expressive and scalable framework. Graph neural networks are used for learning representations for Knowledge graphs and are quite scalable when it comes to probabilistic inference. But no prior rules can be incorporated and it requires significant amount of examples per target in order to converge. On the other hand, MLN are quite powerful for logical reasoning and dealing with noisy data but its inference process is computationally intensive and does not scale. Combining these two frameworks seem to result in a powerful framework which generalizes well to new knowledge graphs, does inference and is able to scale to large entities.

Regarding its contribution, the paper seems to consider a training process which is done using the variational EM algorithm. The variational EM is used to optimize the ELBO term (motivation for this is the intractability of the computing the partition term). In the E-step, they infer the posterior distribution and in the M-step they learn the weights. The integration of variational EM algorithm and MLN has been explored in another work (pLogicNet: Probabilistic Logic Neural Networks for Reasoning), but this paper proposes a new pipeline of tools: MLN, GNN and variational EM which seem to outperform all the existing baseline methods.The paper looks technically sound to me and the evaluations results are delivered neatly, however the flow of the paper makes it a bit difficult to follow sometimes due to many topics covered in it.
Regarding the significance of the paper, it tries to combine logic reasoning and probabilistic inference which is of great interest among the researchers recently. ExpressGNN proves to generalise well and perform accurate inference due to the tunable embeddings added at the GNN.

Overall, the work of this paper seems technically sound but I don’t find the contributions particularly surprising or novel. Along with plogicnet, there have been many extensions and applications of Gnns, and I didn’t find that the paper expands this perspective in any surprising way.


**Experience Assessment:**

I have published one or two papers in this area.

**Review Assessment: Checking Correctness Of Derivations And Theory:**

I assessed the sensibility of the derivations and theory.

**Review Assessment: Checking Correctness Of Experiments:**

I did not assess the experiments.

**Review Assessment: Thoroughness In Paper Reading:**

I read the paper at least twice and used my best judgement in assessing the paper.

---

> ### Author Response · Authors · 2019-11-14
> **Response to Reviewer #1**
>
> Thanks for your comments. We briefly respond to a couple of points as follows.
>
>
> > The integration of variational EM and MLN has been explored in another work pLogicNet.
>
> We have to clarify that our ExpressGNN work was proposed earlier than the pLogicNet. In fact, we have submitted an earlier version of our work to arXiv 15 days before the pLogicNet appeared on arXiv ( https://arxiv.org/abs/1906.08495 ). Due to the ongoing anonymous period, we could not provide the link of our arXiv submission here.
>
>
> > With pLogicNet, the contributions are not surprising or novel.
>
> 1) As claimed above, we proposed the idea of integrating stochastic variational inference and MLN before the pLogicNet work appeared. As a concurrent and later work, pLogicNet also employs variational EM for MLN inference, which should not hurt the originality and novelty of our work.
>
> 2) Compared to pLogicNet, our work employs GNNs to capture the structure knowledge that is implicitly encoded in the knowledge graph. For example, an entity can be affected by its neighborhood entities, which is not modeled in pLogicNet but can be captured by GNNs. Our work models such implicit knowledge encoded in the graph structure to supplement the knowledge from logic formulae, while pLogicNet has no graph structure knowledge and only has a flattened embedding table for all the entities.
>
> 3) Our method is a general framework that can trade-off the model compactness and expressiveness by tuning the dimensionality of the GNN part and the embedding part. Thus, pLogicNet can be viewed as a special case of our work with the embedding part only.
>
> 4) We compared our method with pLogicNet in the experiments. Please refer to Table 3 for the experimental results. Our method achieves significantly better performance than pLogicNet (MRR 0.49 vs 0.33, Hits@10 60.8 vs 52.8) on the FB15K-237 dataset.
>
> We have updated the paper to incorporate the discussions above.

---

### Official Review · AnonReviewer3 · 2019-10-23
**Official Blind Review #3**

**Rating:** 1

**Review:**

The paper proposes to use graph neural networks (GNN) for inference in MLN. The main motivation seems to be that inference in traditional MLN is computationally inefficient. The paper is cryptic about precisely why this is the case. There is some allusion in the introduction as to grounding being exponential in the number of entities and the exponent being related to the number of variables in the clauses of the MLN but this should be more clearly stated (e.g., does inference being exponential in the number of entities hold for lifted BP?). In an effort to speed up inference, the authors propose to use GNN instead. Since GNN expressivity is limited, the authors propose to use entity specific embeddings to increase expressivity. The final ingredient is a mean-field approximation that helps break up the likelihood expression. Experiments are conducted on standard MLN benchmarks (UW-CSE, Kinship, Cora) and link prediction tasks. ExpressGNN achieves a 5-10X speedup compared to HL-MRF. On Cora HL-MRF seems to have run out of memory. On link prediction tasks, ExpressGNN seems to achieve better accuracy but this result is a bit difficult to appreciate since the ExpressGNN can't learn rules and the authors used NeuralLP to learn the rules followed by using ExpressGNN to learn parameters and inference.

Here are the various reasons that prevent me from rating the paper favorably:

- MLNs were proposed in 2006. Statistical relational learning is even older. This is not a paper where the related work section should be delegated to the appendix. The reader will want to know the state of inference and its computational complexity right at the very beginning. Otherwise, its very difficult to read the paper and appreciate the results.

- Recently, a number of papers have been tried to quantify the expressive power of GNNs. MLN is fairly general, being able to incorporate any clause in first-order logic. Does the combination with GNN result in any loss of expressivity? This question deserves an answer. If so, then the speedup isn't free and ExpressGNN would be a special case of MLN, albeit with the advantage of fast inference.

- Why doesn't the paper provide clear inference time complexities to help the reader appreciate the results? At the very least, the paper should provide clear time complexities for each of the baselines.

- There are cheaper incarnations of MLN that the authors should compare against (or provide clear reasons as to why this is not needed). Please see BoostSRL (Khot, T.; Natarajan, S.; Kersting, K.; and Shavlik, J. 2011. Learning Markov logic networks via functional gradient boosting. In ICDM)

**Experience Assessment:**

I have published one or two papers in this area.

**Review Assessment: Checking Correctness Of Derivations And Theory:**

I assessed the sensibility of the derivations and theory.

**Review Assessment: Checking Correctness Of Experiments:**

I assessed the sensibility of the experiments.

**Review Assessment: Thoroughness In Paper Reading:**

I read the paper at least twice and used my best judgement in assessing the paper.

---

> ### Author Response · Authors · 2019-11-14
> **Response to Reviewer #3**
>
> First of all, thank you for your valuable comments. We briefly respond to a couple of points as follows.
>
>
> > Why traditional MLN is computationally inefficient? Provide the inference time complexities.
>
> The computational complexity of probabilistic MLN inference is known to be #P-complete when MLN was proposed [1]. To make it feasible, there are three categories of approximate inference methods: Monte Carlo methods, loopy belief BP, and variational methods [2]. Previous methods (including MCMC, BP, lifted BP) require to fully construct the ground Markov network before performing approximate inference, and the size of the ground Markov network is O(M^d) where M is the number of entities and d is the highest arity of the logic formula. Typically, there are a large number of entities in a practical knowledge graph, making the full grounding infeasible.
>
> With mean-field approximation, our stochastic inference method avoids to fully construct the grounded Markov network, which only requires local grounding of the formulae in each sampled minibatch. Our method has constant time complexity for each sampled minibatch, and the overall time complexity is O(N) where N is the number of iterations. We have compared the inference efficiency on two benchmark datasets. Experimental results reported Fig. 4 show that our method is both more efficient and scalable than traditional MLN inference methods.
>
>
> > Does Lifted BP reduce the computational cost of grounding?
>
> Lifted BP constructs the minimal lifted network via merging the nodes as the first step, and then performs belief propagation on the lifted network to save the computational cost. However, there is no guarantee that the lifted network is much smaller than the ground network. In the worst case, the lifted network can have the same size as the original ground network [2]. Moreover, the construction of the lifted network is also computationally expensive, which is even slower than the construction of the full network as reported in Table 3 of their paper [2]. In fact, our experiments demonstrate that Lifted BP is NOT efficient even on small dataset like UW-CSE and Kinship (please refer to Fig. 4 in our paper), and it certainly cannot scale up to the FB15K-237 dataset.
>
>
> > Why use Neural LP to learn the rules?
>
> The FB15K-237 dataset is not designed for evaluating MLN inference / learning methods, and hence, have no logic formulae provided. Our work focuses on MLN inference and learning with a set of logic formulae, thus we need to generate the rules first. Similarly, recent work [3] uses simple brute-force search to generate the rules for MLN. However, brute-force rule search can be very inefficient on large-scale data. Instead, our method employs Neural LP to efficiently generate the rules. We use the training set only for rule learning, which guarantees that there is no information leakage during the evaluation on the test set.
>
>
> > Why not compare to BoostSRL?
>
> The BoostSRL work uses MC-SAT as the inference method, which has been compared with our work in the experiments. According to the inference time reported in Fig. 4, our method is much more efficient and scalable than MC-SAT.
>
> Moreover, BoostSRL is not directly comparable to our method, since the task is completely different. Our method is designed for MLN inference and rule weight learning with logic rules provided, while BoostSRL was proposed for MLN structure learning, i.e., learning logic rules for MLN. We chose Neural LP instead of this method to generate the rules, since Neural LP has been demonstrated to be effective in rule induction on the Freebase dataset. In the updated paper, we have included BoostSRL as related work to supplement our literature review.
>
>
> > MLN is fairly general, does GNN result in any loss of expressivity?
>
> We have discussed the expressive power of GNNs in our paper in the section titled “Why combine GNN and tunable embeddings”. To make it more clear, in the updated paper, we change the section title to: “Expressive power of GNN as inference network ”. In this section, we have shown an example in Fig. 3 where GNN produces the same embedding for nodes that should be distinguished. We have also formally proved the sufficient and necessary condition to distinguish any non-isomorphic nodes in the knowledge graph. Inspired by this, we augment GNN with additional tunable embeddings to trade-off the compactness and expressiveness of the model.
>
>
> > Related work should appear in the main paper.
>
> Thanks for the suggestion. In the updated paper, we’ve added the related work section right after the introduction to provide a clear background of statistical relational learning and Markov Logic Networks.
>
>
> References
>
> [1] Richardson, Matthew, and Pedro Domingos. “Markov Logic Networks.” Machine Learning.
>
> [2] Singla, Parag, and Pedro M. Domingos. “Lifted First-Order Belief Propagation.” AAAI.
>
> [3] Qu, Meng, and Jian Tang. “Probabilistic Logic Neural Networks for Reasoning.” arXiv.

---

### Public Comment · ~Rainer_Gemulla1 · 2020-03-12
**Test data leakage?**

I had posted the comment below on the Github page accompanying this paper, but perhaps this is a better place for discussion. Quoting:

I've had a look at your recent ICLR20 paper; the results for FB15k-237 are outright amazing! I browsed the source code in this repository to better understand what you do. I stumbled across the following lines in dataset.py:

        for fact in query_ls:
            self.test_fact_ls.append((fact.val, fact.pred_name, tuple(fact.const_ls)))
            self.test_fact_dict[fact.pred_name].add((fact.val, tuple(fact.const_ls)))
            add_ht(fact.pred_name, fact.const_ls, self.ht_dict)

Here query_ls contains the test set facts, and add_ht registers the fact.

If I interpret this correctly, the MLN is constructed as follows. It first adds a variable for each fact r(e1,e2) in the training, validation, and test data. Afterwards, for each such fact, additional variables are (conceptually) added by perturbing e1 or e2: i.e., variables for all facts of form r(e1,?) and r(?,e2) are added as well.

Each of the so-obtained variables is marked as observed (if it appears in the training data) or latent (otherwise).

Is this understanding correct?

The reason I am asking is because such an approach seems to leak validation and test data into training. Why? It's true that the truth values of the validation and test data are not used during training. But: the choice of variables in the MLN already tells the MLN that r(e1,?) and r(?,e2) are sensible queries, and consequently provides information about e1 and e2. That's fine for the training data facts. For validation and test facts, however, it's problematic.

For example, consider a test set fact married_to(JohnDoe, JaneDoe). The mere existence of the variables married_to(JohnDoe, ?) informs the (tuneable) embedding of JohnDoe: it must be a person. Likewise for married_to(?, JaneDoe). That's the first reason for potential leakage. Another reason is that, without any inference or learning, one may "look" at the set of created variables and reduce the set of potential wifes for JohnDoe to the set of persons that have been seen as wifes in the validation or test data. (All facts from the training data are observed so that the corresponding wifes are ruled out.) If so, this would significantly simplify the task.

I'd appreciate if you clarified whether the above description is accurate and, in particular, where I misunderstood the approach.

---

> ### Author Response · Authors · 2020-03-27
> **Clarification on test data usage and inference approach**
>
> Thanks for your interest in our paper. We appreciate your detailed comments. We are truly sorry for being late to respond. Here we clarify our test data usage and details of our inference approach. Hopefully our response clarifies your question, which may also help other readers understand our paper and code.
>
> 1. Potential leakage of validation and test data: 1) We confirm that the truth values of the validation and test data are not used anywhere during the inference and learning process; 2) Our graph neural network (GNN) is built on the knowledge graph, rather than the ground MLN, and we construct the knowledge graph only based on the observed facts in the training data (refer to Fig. 2 for a comparison of the ground MLN and the knowledge graph). That being said, there is no nodes of test set fact in the knowledge graph, thus there is no way of "looking" at the set of created variables and reduce the set of potential answers when updating the tunable embeddings and other trainable parameters of the GNN; 3) For the second part of our E-step objective function in Eq. 6, which is the supervised learning objective for training the GNN, it is only using the observed facts in the training data; 4) When predicting the query married_to(JohnDoe, ?), we consider all the entities in the knowledge graph to replace "?" to construct the test tuples, and predict the probabilities of all the constructed tuples for ranking and computing the evaluation metrics as MRR and Hits@N.
>
> 2. Construction of MLN: The construction of the fully ground Markov Logic Network is computationally infeasible. With the mean-field approximation, we are able to decompose the global expectation over the entire MLN into local expectations over ground formulae. We perform the inference of MLN in a stochastic fashion: we sample mini-batches of ground formulae which may contain both observed and latent variables (facts). For each ground formula in the sampled batch, we take the expectation of the corresponding potential function w.r.t. the posterior of the involved latent variables (first term in Eq. 4), and compute a local sum of entropy using the posterior of the latent variables (second term in Eq. 4). Note that we not only have the facts in training, validation and test data, but also have all the latent variables used in the ground formulae. For example, given there is a logic formula Smoke(x) ∧ Hypertention(x) => Cancer(x), and there's a test fact Cancer(David), then the corresponding variables Smoke(David) and Hypertention(David) in the ground formula will be used for inference and learning, no matter whether they are observed or latent. In summary, we construct MLN based on each ground formula, rather than just adding the facts in training, validation and test data.
>
> 3. How we use test_fact_ls which contains the test set facts: During the inference process, we use test_fact_ls to guide the sampling of ground formulae. There are exponential number of all possible ground formulae, however, most of them are irrelevant to the query facts in the test data. Our sampling strategy is to focus on the ground formulae that contain at least one query fact as the latent variable. We also require that each sampled ground formula should have no truth values yet, i.e., the truth value of ground formula should depend on the truth values of the latent variables in it. This sampling strategy helps find relevant logic formulae that are relevant to the query, so that the inference can be more efficient. Note that there is no guarantee that the sampled formula can derive the correct test fact, since the logic formulae are auto-generated for FB15K-237 by NeuralLP and could be noisy. Our model has no access to the truth values of any latent variables in any ground formulae.
>
> Please kindly let us know if you have any further questions. Thanks.

---

> > ### Public Comment · ~Rainer_Gemulla1 · 2020-03-27
> > **Test data leakage?**
> >
> > Thanks a lot for your feedback so far!
> >
> > I understand that the labels of the test data are not used as labels for their corresponding variables during training. My concern is about your second and third points. As you state in your response, test data is used during training and prediction to create certain variables and factors in the MLN. I am worried that this approach leaks information from the test data.
> >
> > In particular, the performance numbers presented in this paper are far ahead of all numbers I have seen so far (and that haven't been invalidated yet). I'd like to understand whether ExpressGNN really obtains such large improvements, and if so, why that's the case.
> >
> > A quick way to push forward this discussion would be (1) to train ExpressGNN without any access to test data, and (2) to perform evaluation query by query without further access to test data. For example, test triple (s,p,o) has two queries (s,p,?) and (?,p,o), which should be evaluated separately to ensure that there is no leakage.
> >
> > Is this possible? I'd be immediately convinced since (1) ensures that no test data is leaked into training and (2) that no test data is leaked into prediction. If it's not possible to use ExpressGNN like this, why not?
> >
> > As for the discussion, my current understanding is that for a given test triple (s,p,o), the following variables are created:
> >
> > 1. Variable (s,p,o), latent
> > 2. Variables of form (s,p,o') and (s',p,o), latent or observed
> > 3. Variables that occur in the body of (some) rules that have (s,p,o) in their head, latent or observed
> >
> > Is this accurate?
> >
> > If so, both (2) and (3) would leak information.
> >
> > For (2), consider a pathological example: just one test triple (s,p,o), no training or validation data, no rules. Now consider query (s,p,?). Due to the set of variables created in (2), there are many more variables of form (s',p,o)---i.e, with the right answer o---than of form (s',p,o') with o'!=o---i.e., with the wrong answer. Likewise for (?,p,o). One can infer (s,p,o) directly from the set of variables created in (2). In a real setup, the case may not be that pathological, but there is still leakage. That's especially true in conjunction with (3).
> >
> > For (3), the factors introduced for the ground rules are more likely to touch the test triples (the correct answers) than other triples (in particular, the incorrect answers). Again, that's a form of leakage. I understand that "partial grounding" is motivated by performance considerations, but unfortunately it also leaks test data into the resulting distribution. That's also what seems to happen when the sampling strategy "focus[es] on the ground formulae that contain at least one query fact".

---

> > > ### Author Response · Authors · 2020-03-31
> > > **Clarification on test data usage and inference approach**
> > >
> > > Thanks for the follow-up questions. We further clarify these questions as follows.
> > >
> > > 1. We would like to clarify that the training of MLN typically refers to learning the weights of logic formulae, while the inference of MLN is to predict the query (with the current formula weights). In fact, once the MLN is defined with a set of logic formulae and a set of observed facts, the probability of any latent variable (query) is already determined. The reason why we need to update GNN parameters during inference is because the exact inference of MLN is computationally infeasible, and we employ GNN as the variational posterior to perform approximate inference. For the knowledge graph used by GNN, it only contains observed facts and neither latent variables nor test query facts exist in the graph. So when we update the GNN parameters, there are no “created query nodes” in the graph that may leak test data information. We optimize the GNN parameters to make it a better posterior model, so that the variational inference can better approximate the underlying true probability distribution defined by the MLN.
> > >
> > > 2. For each query in the Freebase dataset formed as (s,p,o), we perform inference of (s,p,?) and (?,p,o) sequentially, where we treat the queries as an input stream so that we do keep updating the GNN parameters during the inference. We assume that what you described is with regard to the parameter updating scheme here. Just to confirm with your points, what if we perform the inference of each query independently, i.e., we use the GNN parameters initially learned from supervised data with observed facts only, and perform the inference query by query by updating the GNN parameters always from the initially learned ones, would this way of evaluation rule out the possibility of test data leakage? If so, we can definitely try it out and we’ll update the experimental results here.

---

> > > > ### Public Comment · ~Rainer_Gemulla1 · 2020-04-01
> > > > **Reproducing the experiments without test leakage**
> > > >
> > > > Thank you!
> > > >
> > > > The way you describe parameter updating during prediction seems to be another potential source for potential leakage, but it's not necessarily the only one. I agree with you that a suitable approach is to reproduce the study in a safe way. It's great that you are willing to do this!
> > > >
> > > > To rule out leakage of test data for learning and prediction:
> > > >
> > > > 1. During learning, the test data should not be accessed at all. I had tried to do that by "clearing" the test file in your implementation, but that broke training completely.
> > > >
> > > > To me, the best approach is to update your implementation so that training can be done without accessing the test data file at all. The corresponding model / rule weights should be stored somewhere to use for prediction.
> > > >
> > > > 2. Query-by-query inference is good, but not enough. Again, no access to the test set should arise before seeing a query, and each query should not influence what happens for the next query (in particular, the one for the same triple). For example, any latent variable relevant for a query should be created only once the query is seen and not retained afterwards.
> > > >
> > > > To make sure that there is no leakage in this step, I suggest to provide an CLI that takes a trained model from (1) plus the training data plus a single query (not a triple) and returns the ranked results. Again, a test data file should not be accessed and the trained model must not be changed.
> > > >
> > > > It this feasible to do? Also: in step 2, which variables and factors would be created by your method once seeing a query (say, query (s,p,?)).

---

> > > > > ### Public Comment · ~Rainer_Gemulla1 · 2020-04-01
> > > > > **Additional files in dataset folder**
> > > > >
> > > > > I forgot: in your implementation, there is another file called "facts.txt". It's unclear to me what that file contains. Generally, the safe way is to only access "train.txt" and "valid.txt" during training (step 1), and "train.txt" and a query during prediction (step 2).

---

### Public Comment · ~Rainer_Gemulla1 · 2020-04-27
**Reproducing the experiments without test data leakage**

Thanks again for your willingness to redo the experimental study!

Are there already any updates or is new code available?

---

> ### Author Response · Authors · 2020-04-27
> **Re: Reproducing the experiments without test data leakage**
>
> Thanks for reaching out again. We just started to work on the new experiments. Just to clarify our data setup, the original training data of FB15K-237 is randomly split into "facts.txt" and "train.txt", since we need a subset of training facts to generate first-order logic rules using NeuralLP. So we use "facts.txt" as the knowledge base and "train.txt" as the training data to generate rules.
>
> The crux of our previous discussions seems to be the "clean" way of accessing the test data and updating the GNN parameters during the prediction (inference) phase. As clarified before, we would 1) learn the GNN parameters from supervised KG data with observed facts only; 2) perform query-by-query inference by sampling the query-related ground formulae and updating the GNN parameters in a "sandbox" only for this query, i.e., the update of parameters will not affect the inference of other queries. Please let us know if you have further questions about this new evaluation scheme.
>
> We also noticed that a recent paper published in WWW'20 titled "Probabilistic Logic Graph Attention Networks for Reasoning" is very similar to our work ( https://dl.acm.org/doi/pdf/10.1145/3366424.3391265 ), which reports similar (slightly higher MRR and Hits@10) performance on FB15K-237 compared to our results. Since their paper does not cite our work, we assume they complete the work independently and this may help validate our experimental results.

---

> > ### Public Comment · ~Rainer_Gemulla1 · 2020-04-29
> > **Revised experimental study**
> >
> > Great, thanks! I like your suggestions. Just to clarify (it's probably what you intend to do):
> >
> > 1) Training should not access any test data facts, i.e., it should be possible to do this with the test data file removed.
> >
> > 2) Evaluation should only see individual test queries (but not test triples), as you say. An ideal solution would be to provide an interface for prediction (takes solely a trained model and a query, outputs the ranked results), and use this interface for evaluation.

---

### Public Comment · ~Bin_Dai1 · 2020-05-09
**Question about the tunable embedding**

Hi, thanks for your nice work and the attached code. I tried to reproduce your work using your code. I noticed that you used 127 tunable dimensions and only 1 GNN dimensions in the FB15K-237 experiments (see the last command line in the readme file). Does this setting corresponds to the results in table 3 in the paper?  If so, each layer in the GNN has only 1 dimension, making the network meaningless. I tried to use more GNN dimensions by setting gcn_free_size to a smaller value, but the performance becomes worse. Did I misunderstand something here? Thanks very much.

---

> ### Author Response · Authors · 2020-05-12
> **Re: Question about the tunable embedding**
>
> Thanks for your interest in our work. With regard to the embedding size of GNN / tunable embeddings, we used grid search on the validation set, and the optimal setup is indeed 127-dimensional tunable embeddings plus 1-dimensional GNN embedding, as provided in the default command line. So we also noticed that on the Freebase dataset, higher dimensional GNN is not improving the performance. We assume the reason could be that the supervised learning signal from Freebase dataset is powerful, thus the tunable embedding part is fitting the data pretty well, while the topology of graph is playing less important role in this case. This is reasonable due to the nature of dataset. On the other datasets such as Cora, higher dimensional GNN is not hurting the performance (please refer to Table 2). As discussed in Section 5.1, the inductive GNN embeddings are designed to reduce the number of parameters (more compact model) without hurting the model capacity or expressiveness too much, which is a trade-off between model compactness and expressiveness.

---

> > ### Public Comment · ~Bin_Dai1 · 2020-05-13
> > **Question about the express power**
> >
> > Thanks a lot for your quick response. The clarification is very helpful to me. I have another question about the expressive power of the GNN and the tunable embedding. In your paper, you mentioned that the flattened embedding table proposed in Qu & Tang (2019) is not able to capture the structure knowledge encoded in the knowledge graph. I don't understand why. Here is my thought. Suppose there are $N$ entities and $D$ embedding dimensions. Denote the embedding as $x \in R^{N\times D}$. The optimization space of the tunable embedding is the whole $R^{N\times D}$ space but that of the GNN embedding scheme is just a subset of $R^{N\times D}$ (denoted as $G$). The optimal solution of the GNN embedding is $x^* = \text{\argmin}_{x \in G} L$, where $L$ is the objective while the optimal solution of the tunable embedding is $x^{**} = \text{\argmin}_{x \in R^{N\times D}} L$. It is obvious that $x^{**}$ is at least not worse than $x^*$. The potential reason why $x^*$ will be better than $x^{**}$ could be 1) there are some mechanisms preventing the model to achieve the optimal solution $x^{**}$ in the tunable embedding case, 2) $x^{**}$ is better than $x^*$ on the training set but it generalizes poorly to the test set (if there exists the issue of generalization). Did I make some mistake here?

---

### Public Comment · ~Rainer_Gemulla1 · 2020-06-04
**Any updates?**

Are there any updates or preliminary results that you can share with us? (The GitHub page has not been updated so far.)

---

### Public Comment · ~Xu_Li3 · 2021-01-04
**Concerns about the code in github, seems like using test data in training?**

I have noticed that the main loop of your code use test data during training.
In file:
https://github.com/expressGNN/ExpressGNN/blob/master/data_process/dataset.py
line: 464
you define the function: get_batch_by_q with the argument validation=False
in line: 488-491
the fact_ls is ether equals to valid_fact_ls or test_fact_ls depending on the argument validation

In file:
https://github.com/expressGNN/ExpressGNN/blob/master/main/train.py
line: 69-70
the function is called with the default argument of validation which is False
so that the data you use in the training loop is test_fact_ls right?

by running your code I get the same results (mmr and hist) in your paper.
However, if I changed the data from test_fact_ls to fact_ls(which is built from the training set), the mmr and hits are far lower than the result in the paper.

Is there any misunderstanding of the code or the data stored in test_fact_ls?
Looking forward to your reply.

---

### Decision · Program_Chairs · 2019-12-19

**Decision:**

Accept (Poster)

**Comment:**

This paper is far more borderline than the review scores indicate. The authors certainly did themselves no favours by posting a response so close to the end of the discussion period, but there was sufficient time to consider the responses after this, and it is somewhat disappointing that the reviewers did not engage.

Reviewer 2 states that their only reason for not recommending acceptance is the lack of experiments on more than one KG. The authors point out they have experiments on more than one KG in the paper. From my reading, this is the case. I will consider R2 in favour of the paper in the absence of a response.

Reviewer 3 gives a fairly clear initial review which states the main reasons they do not recommend acceptance. While not an expert on the topic of GNNs, I have enough of a technical understanding to deem that the detailed response from the authors to each of the points does address these concerns. In the absence of a response from the reviewer, it is difficult to ascertain whether they would agree, but I will lean towards assuming they are satisfied.

Reviewer 1 gives a positive sounding review, with as main criticism "Overall, the work of this paper seems technically sound but I don’t find the contributions particularly surprising or novel. Along with plogicnet, there have been many extensions and applications of Gnns, and I didn’t find that the paper expands this perspective in any surprising way." This statement is simply re-asserted after the author response. I find this style of review entirely inappropriate and unfair: it is not a the role of a good scientific publication to "surprise". If it is technically sound, and in an area that the reviewer admits generates interest from reviewers, vague weasel words do not a reason for rejection make.

I recommend acceptance.